# A distinctive family of L,D-transpeptidases catalyzing L-Ala-mDAP crosslinks in Alpha- and Betaproteobacteria

Akbar Espaillat[1,6,7], Laura Alvarez[1,7], Gabriel Torrens[1,7], Josy ter Beek[2,3], Vega Miguel-Ruano[4], Oihane Irazoki[1], Federico Gago[5], Juan A. Hermoso[4], Ronnie P-A. Berntsson[2,3] & Felipe Cava[1]✉

The bacterial cell-wall peptidoglycan is made of glycan strands crosslinked by short peptide stems. Crosslinks are catalyzed by DD-transpeptidases (4,3-crosslinks) and LD-transpeptidases (3,3-crosslinks). However, recent research on non-model species has revealed novel crosslink types, suggesting the existence of uncharacterized enzymes. Here, we identify an LD-transpeptidase, LDT$_{Go}$, that generates 1,3-crosslinks in the acetic-acid bacterium *Gluconobacter oxydans*. LDT$_{Go}$-like proteins are found in Alpha- and Betaproteobacteria lacking LD3,3-transpeptidases. In contrast with the strict specificity of typical LD- and DD-transpeptidases, LDT$_{Go}$ can use non-terminal amino acid moieties for crosslinking. A high-resolution crystal structure of LDT$_{Go}$ reveals unique features when compared to LD3,3-transpeptidases, including a proline-rich region that appears to limit substrate access, and a cavity accommodating both glycan chain and peptide stem from donor muropeptides. Finally, we show that DD-crosslink turnover is involved in supplying the necessary substrate for LD1,3-transpeptidation. This phenomenon underscores the interplay between distinct crosslinking mechanisms in maintaining cell wall integrity in *G. oxydans*.

Most bacteria are protected by an extracytoplasmic cell wall that mainly consists of a net-like structure, named the peptidoglycan[1]. Peptidoglycan, also known as the murein *sacculus* (Latin for "small bag"), functions as an exoskeleton that protects bacterial cells from bursting due to their high internal turgor pressure[2]. Based on its organization, bacteria can be defined as Gram-negative and Gram-positives[3]. Gram-negative bacteria have a thin peptidoglycan monolayer confined to the periplasmic space, a cellular compartment between the cytoplasmic and the outer membranes, while Gram-positives instead have a thick, multi-layered peptidoglycan outside of the cytoplasmic membrane and no outer membrane[4]. At the compositional level, peptidoglycan is a heteropolymer made of glycan strands crosslinked by short peptide stems. The canonical peptidoglycan subunit (i.e., muropeptide) consists of the disaccharide pentapeptide, N-acetylglucosamine (NAG) β(1→4) N-acetylmuramic acid (NAM)-L-alanine[1]-(γ)-D-glutamate[2]-(diamino acid[3])-D-alanine[4]-D-alanine[5] (abbreviated here as M5, for monomer disaccharide-pentapeptide). Usually, the diamino acid in the third position of the murein peptide stems is meso-diaminopimelic acid (mDAP[3]) in Gram-negatives and L-Lys in Gram-positives[1]. However, various species

[1]Department of Molecular Biology and Laboratory for Molecular Infection Medicine Sweden, Umeå Centre for Microbial Research, SciLifeLab, Umeå University, Umeå, Sweden. [2]Department of Medical Biochemistry and Biophysics, Umeå University, Umeå, Sweden. [3]Wallenberg Centre for Molecular Medicine, Umeå University, Umeå, Sweden. [4]Department of Crystallography and Structural Biology, Institute of Physical Chemistry "Blas Cabrera", CSIC, Madrid, Spain. [5]Department of Biomedical Sciences & IQM-CSIC Associate Unit, School of Medicine and Health Sciences, University of Alcalá, E-28805 Madrid, Alcalá de Henares, Spain. [6]Present address: Chr. Hansen A/S, Microbial Physiology, R&D, 2970 Hoersholm, Denmark. [7]These authors contributed equally: Akbar Espaillat, Laura Alvarez, Gabriel Torrens. ✉e-mail: felipe.cava@umu.se

have additional peptidoglycan chemical changes that typically enable adaptation of the cell wall to specific environmental challenges[5], e.g., substitution of the D-alanine at fifth position (D-Ala[5]) of the peptide moiety by D-lactate in vancomycin-resistant strains[6].

Peptidoglycan synthesis requires the coordinated action of synthetic and degradative enzymes that catalyze the insertion of new material into the pre-existing murein sacculus, thereby enabling sacculi expansion to support cell growth. Murein polymerization requires transglycosylase (TGase) and transpeptidase (TPase) activities. TGase enzymes include the bifunctional class A penicillin-binding proteins (PBPs), the shape, elongation, division, and sporulation proteins (SEDS), and the monofunctional glycosyl transferases[7]. Peptidoglycan transpeptidation (i.e., crosslinking) is mainly catalyzed by class A and monofunctional class B PBPs, also called DD-TPases. However, many bacteria encode alternative TPases known as LD-TPases[8] that do not share sequence homology with PBPs. They present a YkuD-like domain (PFAM 03734) that includes a cysteine as the catalytic nucleophile instead of the conserved serine in PBPs[8]. PBPs cleave the terminal peptide bond between the fourth and fifth amino acid of the donor pentapeptide (D-Ala[4]-D-Ala[5]) to form a new peptide bond connecting the D-Ala[4] of the donor muropeptide with the D-chiral center of the mDAP[3] from an adjacent acceptor muropeptide, thereby forming a 4,3- or DD-crosslink. Contrary to PBPs, LD-TPases cleave between mDAP[3]-D-Ala[4] in the donor tetrapeptide and use the energy to form a crosslink between the L- and the D-center of two adjacent mDAP[3] residues, thereby producing a 3,3- or LD-crosslink.

The activity of PBPs is vital for building a viable cell wall, and as a result, most bacteria carry one or more indispensable PBPs. However, LD-transpeptidation is not essential for survival, but is important for a number of processes such as chemical editing of peptidoglycan with non-canonical D-amino acids (NCDAA)[9], tethering of outer membrane proteins to the peptidoglycan[10,11], toxin secretion[12], lipopolysaccharide translocation[13], antibiotic resistance[14], and polar growth[15].

We previously reported that Acetobacteraceae, a family of Gram-negative bacteria belonging to the class Alphaproteobacteria, presents a unique peptidoglycan chemical structure, which includes a novel 1,3-type LD-crosslink between L-Ala[1]-DAP[3] [16]. These microbes, also known as acetic acid bacteria, include ubiquitous strictly aerobic mesophilic species that produce acetic acid during oxidative fermentation and play important roles in the food industry.

Here, using *Gluconobacter oxydans* as a model organism, we have identified the LD-TPase, LDT$_{Go}$, responsible for the creation of these unconventional 1,3 peptidoglycan crosslinks. We conducted an in-depth exploration of its structural characteristics, biochemical properties, and relevance in biological contexts. We show that LDT$_{Go}$ has distinctive structural features with respect to 3,3 crosslinking enzymes, which includes the active site donor cavity and an N-proximal region that likely has a regulatory function on its enzymatic activity. Importantly, we demonstrate that contrary to what is seen in other crosslinking enzymes, LD1,3-TPases have transferase activity of non-terminal (endo) peptide bonds, thus enabling these enzymes to use multiple donor muropeptides. We further show that although LDT$_{Go}$ is constitutively expressed, LD1,3-crosslinking levels increase in stationary phase and we demonstrate this regulation depends on the DD-endopeptidase (DD-EPase) activity of PBP7, which controls substrate availability for LDT$_{Go}$. This growth phase-dependent crosstalk between DD- and LD-crosslinking types supports peptidoglycan crosslink homeostasis in *G. oxydans*. Inactivation of LDT$_{Go}$ sensitizes *G. oxydans* to cell envelope stresses, which cannot be complemented by canonical 3,3-type LD-crosslinks. Conservation of LDT$_{Go}$-like enzymes within the Alpha- and Betaproteobacteria suggests an important role for this family of enzymes in shaping the cell wall of these organisms to adapt to challenging conditions.

## Results

### Identification of an LD1,3-transpeptidase in *G. oxydans*

We previously found that the peptidoglycan of acetic acid bacteria (Acetobacteria) exhibits an unusual type of crosslink between the L-Ala at position 1 of the donor muropeptide stem peptide and the D-chiral center of mDAP at position 3 of the acceptor muropeptide (L-Ala[1]-DAP[3]) [16] (Fig. 1A). We reasoned that an LD-TPase might catalyze this crosslinking reaction but found no homologs of the canonical LD-TPases that catalyze 3,3-crosslinks. As LD-TPases typically have a characteristic YkuD domain[8], we then searched for genes encoding YkuD-containing proteins in the genome of *G. oxydans* and found *gox2269* and *gox1074* (Fig. 1B). While the muropeptide profile of a clean deletion mutant strain Δ*gox2269* was identical to the wild-type (Supplementary Fig. 1), that of Δ*gox1074* exhibited a notable absence of 1,3-crosslinks, which were fully restored through ectopic complementation (Fig. 1C), demonstrating Gox1074 plays an essential role in peptidoglycan LD1,3-crosslinking in *G. oxydans*.

The YkuD domain of Gox1074 includes the putative catalytic cysteine (C264) and histidine (H245) which are conserved throughout LD3,3-TPases (Fig. 1D). To assess if these residues are essential for Gox1074 function, we complemented the Δ*gox1074* strain with alleles in which these residues were replaced by alanine (C264A and H245A strains) and analyzed their peptidoglycan by UPLC-UV. Like Δ*gox1074*, the muropeptide profile of these strains completely lacked LD1,3-crosslinks, supporting that C264 and H245 are critical residues for Gox1074 activity (Fig. 1C).

SignalP 6.0[17] predicted Gox1074 has an N-terminal Sec/SPII signal peptide typical of prokaryotic lipoproteins that includes the consensus sequence [LVI][ASTVI][GAS]C (Supplementary Fig. 2A). By immuno-detection, we found that Gox1074 is associated to the membrane protein fraction (Supplementary Fig. 2B). For other TPases, it has been demonstrated that their association with the membrane facilitates critical protein-protein interactions for catalysis and localization[18,19]. To investigate whether this is the case for Gox1074, we produced two additional Δ*gox1074* strains: one expressing *gox1074* without its lipoprotein signal peptide (ΔSP) and one expressing *gox1074* that instead had the signal peptide from YcbB$_{Ec}$ (SP$_{YcbB}$), which is predicted to localize to the periplasm and get cleaved in *E. coli*[20] (Supplementary Fig. 2C). Our results suggest that while Gox1074 needs to be exported to the periplasm to be functional, it does not need to be anchored to the membrane (Supplementary Fig. 2D, E). Based on these results we renamed Gox1074 as LDT$_{Go}$, for LDT of *G. oxydans*, and termed LDT$_{Go}$-like enzymes LD1,3-TPases to distinguish them from the LD-TPases producing 3,3 crosslinks (LD3,3-TPases).

### Conservation of LD1,3-TPases

To analyze the conservation of LD1,3-TPases, we built a phylogenetic tree based on sequence similarity. We found that LDT$_{Go}$ homologs are encoded in Alpha- and Betaproteobacteria, particularly among the Acetobacteraceae (e.g., *Acetobacter pasteurianus*) and Burkholderiaceae families (e.g., *Burkholderia cenocepacia*) but also in some Comamonadaceae (e.g., *Rhoderax lacus*), Alcaligenaceae (e.g., *Achromobacter denitrificans*) and Oxalobacteraceae (e.g., *Herminiimonas fonticola*). Outside of these phyla, orthologs can also be found in some Desulfovibrionaceae species (e.g., *Solidesulfovibrio carbinoliphilus*) (Fig. 2A). Sequence alignment of representative LDT$_{Go}$-like proteins belonging to the above indicated taxa showed these proteins maintain the catalytic YkuD domain but not the lipobox SP (Fig. 2B). Interestingly, LD1,3- and LD3,3-TPases appear to be mutually exclusive as homologs of LDT$_{Go}$ and YcbB$_{Ec}$ are not encoded in the same species (Fig. 2A).

To assess whether these predicted LDT$_{Go}$-like proteins are authentic LD1,3-TPase enzymes, we expressed 2 orthologs, from *A. pasteurianus* and *Burkholderia cenocepacea*, in *E. coli* (which lacks 1,3-

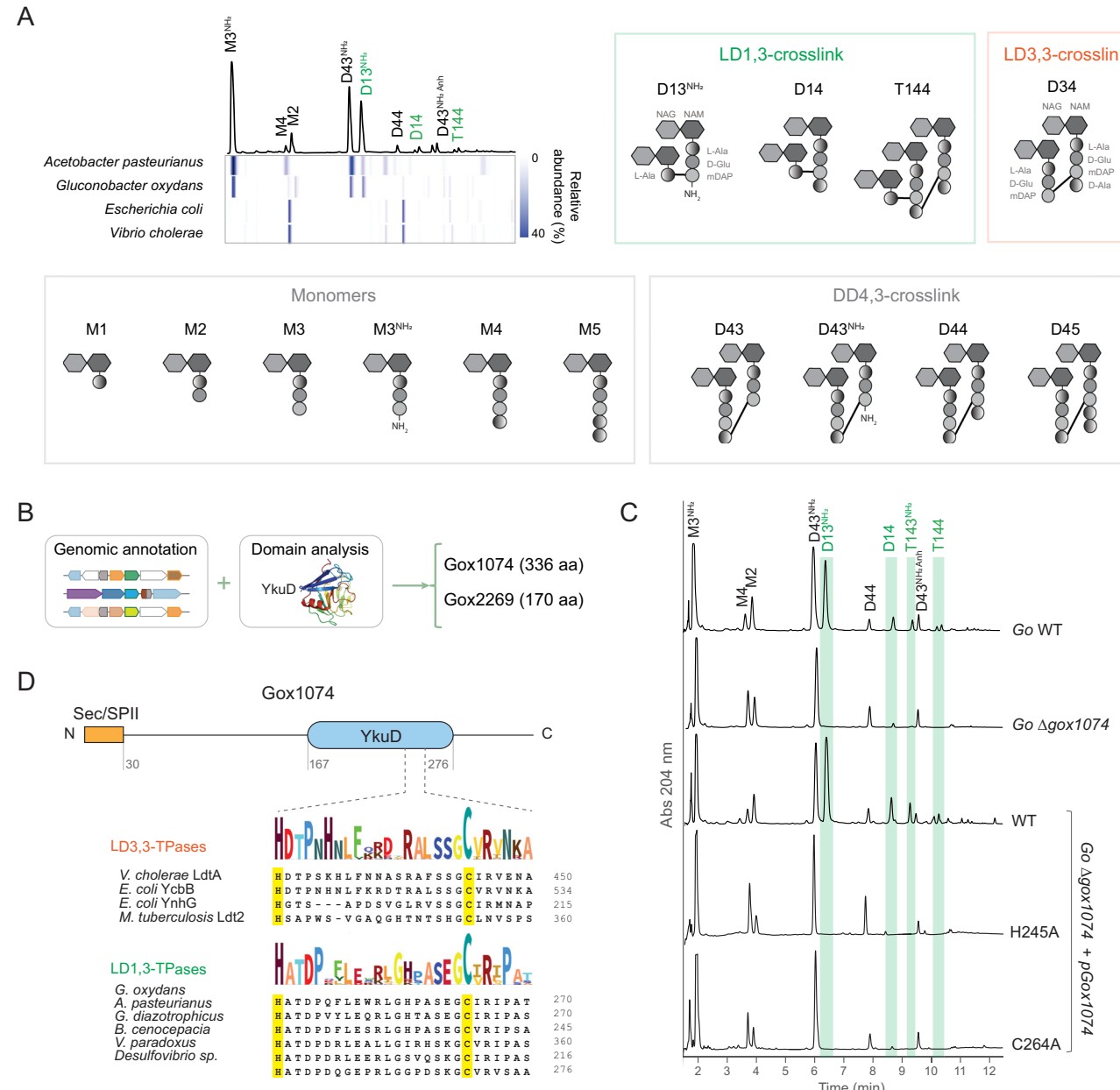

**Fig. 1 | Identification of the enzyme catalyzing LD1,3-crosslink. A** Representative UV chromatogram of *Gluconobacter oxydans* peptidoglycan profile in stationary phase. LD1,3-crosslinked muropeptides are indicated in green. Underneath, heatmap representing the relative abundance of each muropeptide in representative bacteria species from the Acetobacteraceae family (*Acetobacter pasteurianus* and *G. oxydans*) and model organisms *Escherichia coli* and *Vibrio cholerae*. Schematic structures and nomenclature of the main muropeptides and crosslink type are shown. **B** Scheme of the in silico search of YkuD-domain containing enzyme candidates responsible for LD1,3-crosslinking. **C** Representative UV chromatograms of *G. oxydans* (*Go*) wild-type (WT), *Δgox1074* and *Δgox1074* pGox1074 complemented strains. LD1,3-crosslinked muropeptides are highlighted in green. **D** Domain analysis of Gox1074. Details of the LDT conserved motif in the YkuD domain including the catalytic Cys and His residues highlighted in yellow.

crosslinks) and subsequently analyzed the peptidoglycan structure. Both proteins produced LD1,3-crosslinks in *E. coli*; however, the *B. cenocepacia* protein (referred to as LDT$_{Bcn}$, BCAM2463) generated much higher levels of the crosslink compared to both LDT$_{Go}$ and its counterpart from *A. pasteurianus* (LDT$_{Ap}$, NBRC3222_0766). As for LDT$_{Go}$, LDT$_{Bcn}$ activity was abolished by introducing a point mutation that changed the presumed catalytic cysteine to an alanine (C354A) (Fig. 2C) and successfully complemented *G. oxydans* *Δldt$_{Go}$* (Fig. 2D). Interestingly, although expressing these enzymes in *E. coli* leads to substantial LD1,3-crosslinking, these muropeptide levels are comparatively scarce in their native species (Fig. 1D, Supplementary Fig. 3[16]), thereby suggesting the existence of species-specific

regulatory mechanisms controlling the activity or expression of these enzymes.

Collectively, these findings indicate that LD1,3-TPase enzymes exhibit a high degree of conservation across various families within the Alpha- and Betaproteobacteria that lack LD3,3-TPases.

## LD1,3-TPase donor substrate specificity

To make LD-crosslinked dimers, LD3,3-TPases use disaccharide tetrapeptides (M4) as donor muropeptide[21]. By analogy, we reasoned that the disaccharide-dipeptide (M2) should serve as a donor in the production of LD1,3-crosslinked dimers. However, this seemed unlikely due to the low abundance of this muropeptide (5.4%) in *E. coli*

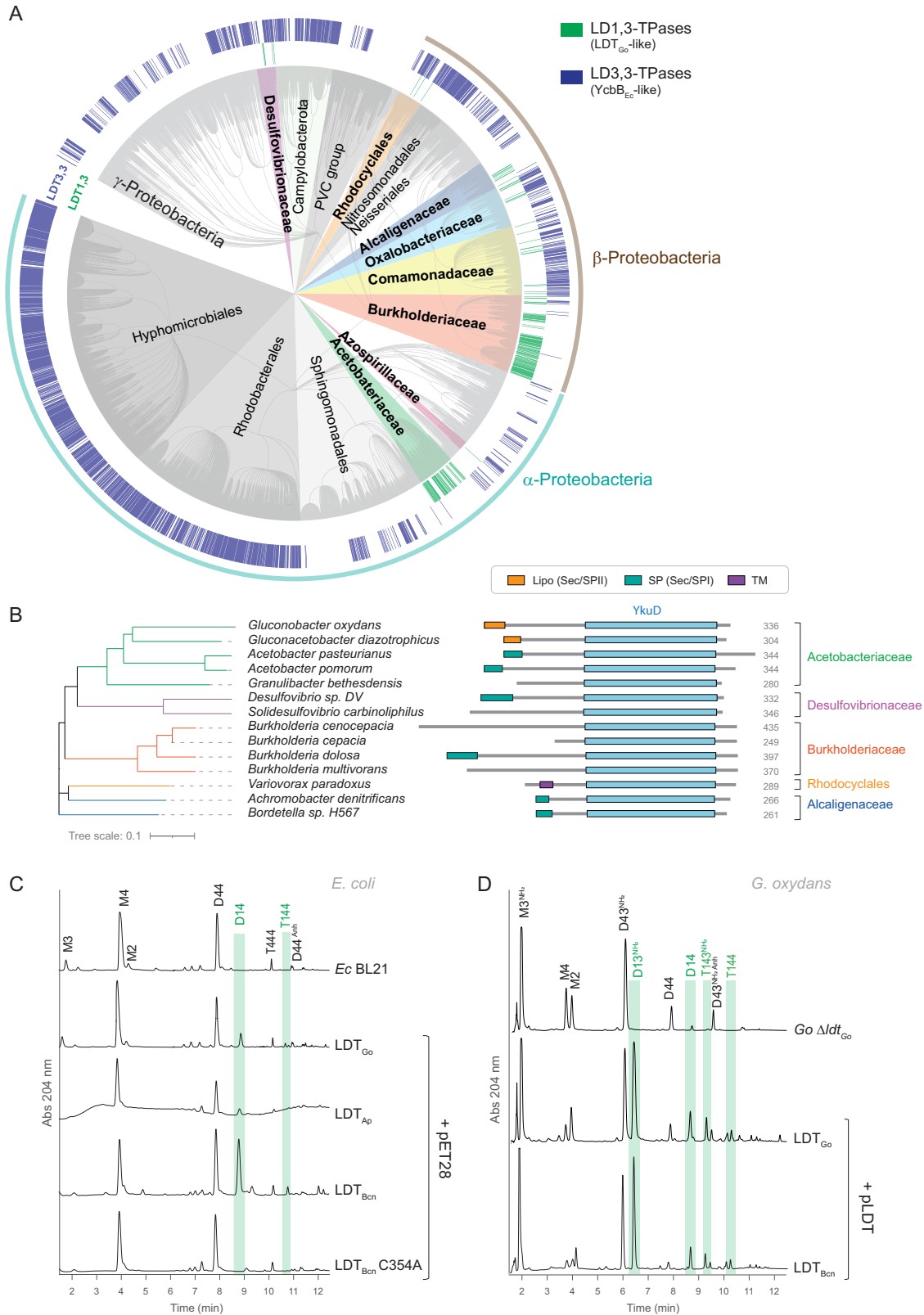

peptidoglycan, particularly in contrast to the substantial production of the 1,3-crosslinked dimer D14 (20.5%) during the overexpression of $LDT_{Bcn}$ in this bacterium (Fig. 2C). Given the significant reduction (34%) in M4 levels upon the expression of $LDT_{Bcn}$ in *E. coli* (Fig. 2C), we postulated that M4 rather than M2 could serve as the donor in LD1,3-transpeptidation reactions.

To investigate this hypothesis, we opted to use $LDT_{Bcn}$ over $LDT_{Go}$ due to its superior activity in *E. coli* (Fig. 2C) and performed in vitro LD1,3-TPase assays on *V. cholerae* sacculi, which lacks LD1,3-crosslinks and only contains minimal levels of M2. We successfully detected D14 and the trimer T144, confirming $LDT_{Bcn}$ was active in vitro (Fig. 3A, B, Supplementary Fig. 4). As for LD3,3-TPases[22,23],

**Fig. 2 | Conservation of LDT_Go. A** Phylogenetic tree showing the conservation and distribution of LD-transpeptidases. In green, LDT_Go-like proteins (LD1,3-TPases) and in blue, homologs to YcbB_Ec (LD3,3-TPases). Homology was assessed by BLAST against the NCBI and OrthoDBv11 databases. **B** Comparison of the domains in representative LDT_Go orthologues from diverse bacterial Families: WP_011252635.1 (*Gluconobacter oxydans*), WP_041249327.1 (*Gluconacetobacter diazotrophicus*), WP_124305792.1 (*Acetobacter pasteurianus*), WP_099541073.1 (*Acetobacter pomorum*), WP_253736052.1 (*Granulibacter bethesdensis*), WP_254845249.1 (*Desulfovibrio sp. DV*), EHJ47292.1 (*Solidesulfovibrio carbiniliphilus*), B4EIM9_BURCJ (*Burkholderia cenocepacia*), QFS41266.1 (*Burkholderia cepacian*), WP_244096448.1

(*Burkholderia dolosa*), AYZ63023.1 (*Burkholderia multivorans*), WP_215249195.1 (*Variovorax paradoxus*), ALX84167.1 (*Achromobacter denitrificans*), AOB33672.1 (*Bordetella sp. HS67*). The presence of a signal peptide is indicated. Signal peptides and transmembrane domains are predicted using SignalP 6.0. **C** UV muropeptide profiles of the heterologous expression of LDT_Go and its homologs from *Acetobacter pasteurianus* (LDT_Ap), *Burkholderia cenocepacia* (LDT_Bcn) and a catalytically inactive mutant (LDT_Bcn C354A) in *E. coli* BL21. LD1,3-crosslinked muropeptides are highlighted in green. **D** UV muropeptide profiles of *G. oxydans* (Go) Δ*ldt_Go* mutant and complemented derivatives expressing the LDT_Go and LDT_Bcn. LD1,3-crosslinked muropeptides are highlighted in green.

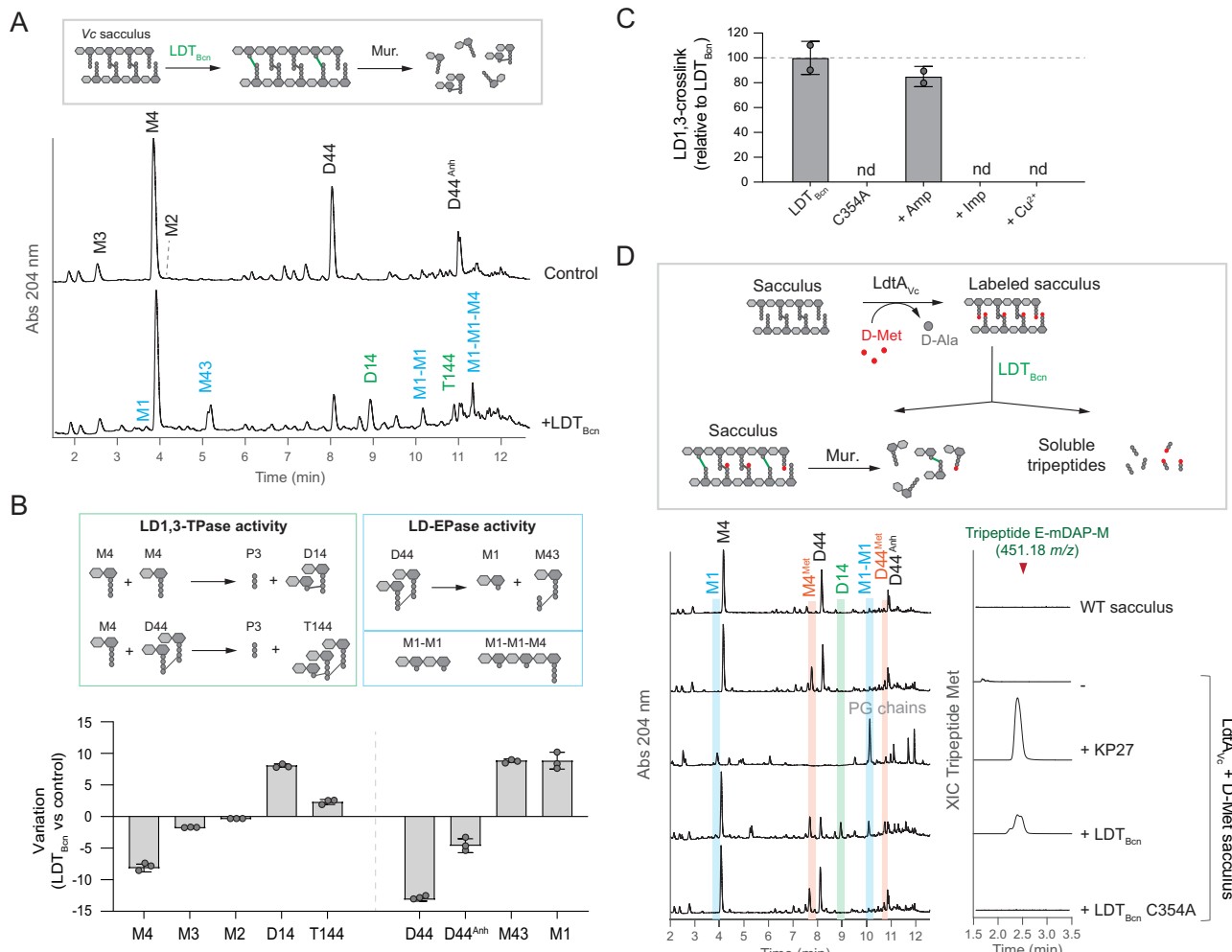

**Fig. 3 | LD1,3-crosslinking activity. A** In vitro activity assays of LDT_Bcn on M4-rich peptidoglycan sacculi from *V. cholerae* vs control (no enzyme added). LD1,3-crosslinked muropeptides are labeled in green and the products of LDT_Bcn endopeptidase activity are labeled in blue. **B** Muropeptide quantifications from panel (**A**). Variation is calculated as the difference in relative molar abundance of the muropeptide in the LDT_Bcn treated reaction minus control reaction in the in vitro assays. Relative molar abundances were calculated as the percentage of the peak area of a muropeptide, divided by its molecular weight, compared to the sum of peak areas in the chromatogram. **C** Effect of Ampicillin 100 μg/ml (Amp), Imipenem 100 μg/ml (Imp) and copper 1 mM (Cu²⁺) on the in vitro assays shown in panel (**A**).

nd: not detected. **D** Scheme and in vitro activity assays of LDT_Bcn on M4-rich peptidoglycan sacculi previously modified with incorporated D-Met at the terminal position of the tetrapeptides. KP27 endopeptidase cleaving between L-Ala[1] and D-Glu[2] was used as control. Left side, UV muropeptide profiles of the mutanolysin-digested insoluble pellets showing the presence of D-Met modified muropeptides (in red), endopeptidase products (M1 and M1-M1, in blue) and LD1,3-crosslinked muropeptides (in green). The MS extracted ion chromatogram (XIC) trace of the D-Met containing tripeptide (E-mDAP-M) in the soluble fraction is shown on the right side. Error bars in graphs (**B**, **C**) represent standard deviation from mean. Source data for (**B**, **C**) are provided as a Source Data file.

synthesis of LD1,3-crosslinks was fully inhibited by imipenem and Cu²⁺ but not by ampicillin (Fig. 3C, Supplementary Fig. 4). Notably, the formation of these LD1,3-crosslinked species was accompanied by a substantial decrease in M4, the main monomer (Fig. 3A, B). In addition, we observed a significant reduction in D44 and the simultaneous generation of M43 (a disaccharide-tetrapeptide crosslinked to a

tripeptide, Supplementary Table 5) and M1, indicating that this enzyme also exhibits a notable endopeptidase activity in vitro, particularly on DD-crosslinked tetrapeptides (Fig. 3A, B). These results support our hypothesis that LDT_Bcn utilizes tetrapeptides as a substrate and further suggest that like for classical L,D-transpeptidases, LDT_Bcn forms an acyl intermediate that can be hydrolyzed leading to

LD-EPase activity or react with an acyl acceptor (e.g., M4) leading to LD-TPase activity.

We then reasoned that if LDT$_{Bcn}$ uses M4 as a donor muropeptide, we should observe the release of the tripeptide D-Glu-mDAP-D-Ala when D14 is generated (Fig. 3B). To investigate this hypothesis, we used the phage endopeptidase KP27[24] as control, which cleaves peptidoglycan between L-Ala[1]-D-Glu[2] and thus releases the same tripeptide (Supplementary Fig. 5A). However, even though KP27 produces high levels of M1 muropeptide and significantly reduces M4 levels, we were unable to detect D-Glu-mDAP-D-Ala, likely because it elutes at the solvent front (Supplementary Fig. 5B). To address this challenge, we initially substituted the terminal D-Ala[4] by D-Met, using the LD-TPase LdtA[9] (Fig. 3D) with the reasoning that the delayed elution of the more hydrophobic D-Met-containing tripeptide (D-Glu-mDAP-D-Met) would allow its detection. This time, we successfully identified both M1 and D-Glu-mDAP-D-Met in the peptidoglycan samples treated with KP27 (Fig. 3D). When D-Met-labeled peptidoglycan was treated with LDT$_{Bcn}$, it generated D14 dimers and M1 muropeptides, and released the D-Glu-mDAP-D-Met tripeptide, thereby confirming this LD1,3-TPase can use M4 as the donor muropeptide in transpeptidase and endopeptidase reactions (Fig. 3D).

To explore whether M3, like M4, can also serve as the donor muropeptide, we conducted comparable in vitro LD1,3-transpeptidation assays. However, this time, we applied these assays to sacculi in which the tetrapeptides were enzymatically trimmed to tripeptides by an LD-carboxypeptidase (LdcA) (Supplementary Fig. 6A)[25]. As expected, LDT$_{Bcn}$ activity on this "M3-enriched" sacculi produced LD-crosslinked dimers (D13) that were on par with those measured when using "M4-rich" peptidoglycan as substrate (Supplementary Fig. 6B, C). While LdcA efficiently trimmed monomer tetrapeptides, it showed poor activity on D44, hence we detected again the LDT$_{Bcn}$ endopeptidase products M43 and M1 as before (Supplementary Fig. 6B, C). Once more, no changes in the M2 levels were observed in the samples treated with LDT$_{Bcn}$, reinforcing the notion that M2 is not the favored donor muropeptide (Supplementary Fig. 6B, C). Interestingly, no LD1,3-crosslinked muropeptides were detected when we used a mutanolysin-digested peptidoglycan or purified muropeptides. This suggests a preference of LDT$_{Bcn}$ for larger peptidoglycan structures, such as short chains, rather than individual muropeptides.

In summary, these results conclusively establish that LD1,3-TPases represent a distinct category of transpeptidases which conduct L-Ala[1] and D-Glu[2] endopeptidation on donor muropeptides with diverse peptide stem lengths, ultimately enabling the formation of LD1,3-crosslinks.

## LD1,3-TPase performs D,L-amino acid exchange reactions

Additional differences in the catalysis of LD1,3- and LD3,3-TPases were observed in their ability to perform amino acid exchange reactions. As LD3,3-TPases can exchange the D-Ala[4] of the peptide stems for non-canonical D-amino acids (NCDAA, D-AA other than the canonical D-Ala and D-Glu), fluorescent D-amino acids (FDAA) or clickable DAA[9,26,27], we wondered whether LD1,3-TPases could also perform a similar D-amino acid exchange reaction (Fig. 4A). However, no incorporation of D-Met or the FDAA HADA was detected in G. oxydans wild-type cells compared to the control strain in which ldt$_{Go}$ was chromosomally replaced with ycbB$_{Ec}$ (ldt$_{Go}$::ycbB) (Fig. 4B, C, Supplementary Fig. 7). Interestingly, using UPLC-MS we detected trace amounts of several M2 ions in the peptidoglycan of G. oxydans that were absent in the Δldt$_{Go}$ mutant strain, corresponding to dipeptides with the D-Glu[2] replaced with Phe and Trp (Fig. 4D, Supplementary Figs. 8 and 9). These M2$^{Phe}$ and M2$^{Trp}$ species were present also in E. coli overexpressing the LDT$_{Bcn}$ wild-type but not its catalytically inactive C354A mutant version (Supplementary Fig. 9B). Notably, we also detected M2$^{Gln}$ but its production is independent of LDT$_{Go}$, as it is found in G. oxydans Δldt$_{Go}$ and in E. coli overexpressing the LDT$_{Bcn}$ wild-type or LDT$_{Bcn}$ C354A (Fig. 4D,

Supplementary Fig. 9). As neither G. oxydans nor E. coli encode homologs of the broad-spectrum racemase BsrV[28,29], the enzymes that typically produce NCDAAs, we reasoned that Phe and Trp were likely L-amino acids. To test this hypothesis, we first treated G. oxydans sacculi (which contain significant levels of M2) with LDT$_{Bcn}$ in vitro in the presence and absence of L-Phe and D-Phe. However, we found no significant changes in the M2$^{Phe}$ levels between these samples, indicating a lack of this activity under our in vitro conditions (Supplementary Fig. 10). Therefore, we tried growing G. oxydans wild-type and Δldt$_{Go}$ strains in cultures supplemented with 10 mM of L-Phe or D-Phe and monitored the generation of the M2$^{Phe}$ muropeptide. Our results detected increased levels of M2$^{Phe}$ (2.5-fold) only in the wild-type cultures supplemented with L-Phe, suggesting that in addition to its LD1,3-transpeptidase and endopeptidase activities, LDT$_{Go}$ has D,L-amino acid exchange activity, e.g., D-Glu[2] for L-Phe (Fig. 4E).

## Three-dimensional structure of LDT$_{Go}$

To understand the distinctive structural and catalytic characteristics of LD1,3-TPase enzymes in relation to their LD3,3-TPase counterparts, we determined the crystal structure of a functional soluble LDT$_{Go}$ variant at a resolution of 1.7 Å (PDB ID: 8QZG, Fig. 5A, Supplementary Fig. 11). The structure was solved by the molecular replacement method by using the catalytic domain as predicted by AlphaFold2 (AF2). It is worth mentioning that the region comprising residues 56–86 (referred later as the "belt") was not properly predicted by the AF2 model and was manually modeled into the experimental electron density map (see Methods for further details). The electron density allowed us to build a model for residues 54–331, except for residues 201–215 that are likely in a flexible region and correspond to the capping loop, a subdomain that has been hypothesized to assist in binding peptidoglycan substrates in YcbB$_{Ec}$[20]. Comparison of the structures of LDT$_{Go}$ and YcbB$_{Ec}$ (PDB ID: 6NTW), revealed that while both proteins present YkuD domains with superimposed catalytic Cys and His residues, the overall structure of the domain is rather different (Fig. 5B). One striking change is in the capping loop, which is larger in YcbB$_{Ec}$ and smaller and partially disordered in LDT$_{Go}$. A second major difference is in the size and conformation of the β sheet and loops surrounding the catalytic center (Fig. 5B). As detailed below, these structural differences are essential to understanding the unique catalytic properties exhibited by LDT$_{Go}$.

LDT$_{Go}$ contains an elongated and unstructured proline-rich N-proximal region, spanning until residue 86, which we will refer to as the "belt" for reasons we explain below. Although this region demonstrates variability in its amino acid sequence, it is broadly conserved among LD1,3-TPases, while being conspicuously absent in LD3,3-TPases (Supplementary Fig. 12). In our crystal structure, the belt wraps around the protein, forming up to 11 hydrogen bonds, various van der Waals interactions and inserts into the catalytic cavity of the YkuD domain (Fig. 5C, Supplementary Table 1). It is noteworthy that Asn61 from the belt is making hydrogen bonds with two residues located at both sides of the catalytic Cys264 (Fig. 5C), the catalytic His245 and the Glu262 that, as described below, is part of the acidic patch close to the acyl-donor site. Thus, the obtained crystal structure of LDT$_{Go}$ likely represents a self-inhibited state in which the belt occludes the active site from the peptidoglycan and blocks the catalytic residues.

In the crystal structure, the capping loop is disordered and the L$_{247-252}$ presents a conformation in which the Phe251 traps Arg266 from the active site through a cation-π interaction (Fig. 5D, middle panel). We ran a molecular dynamics (MD) simulation with the crystal structure which showed that the belt and the capping loop had the largest variation during a 350 ns simulation (Supplementary Fig. 13A, B, Supplementary Movie 1). Furthermore, MD simulations in which we removed the belt in silico generated a model that was remarkably similar to that predicted by AF2 for LDT$_{Go}$, where the belt does not encircle the protein, allowing the active site to remain unobstructed (Supplementary Fig. 13C). In this model, the capping

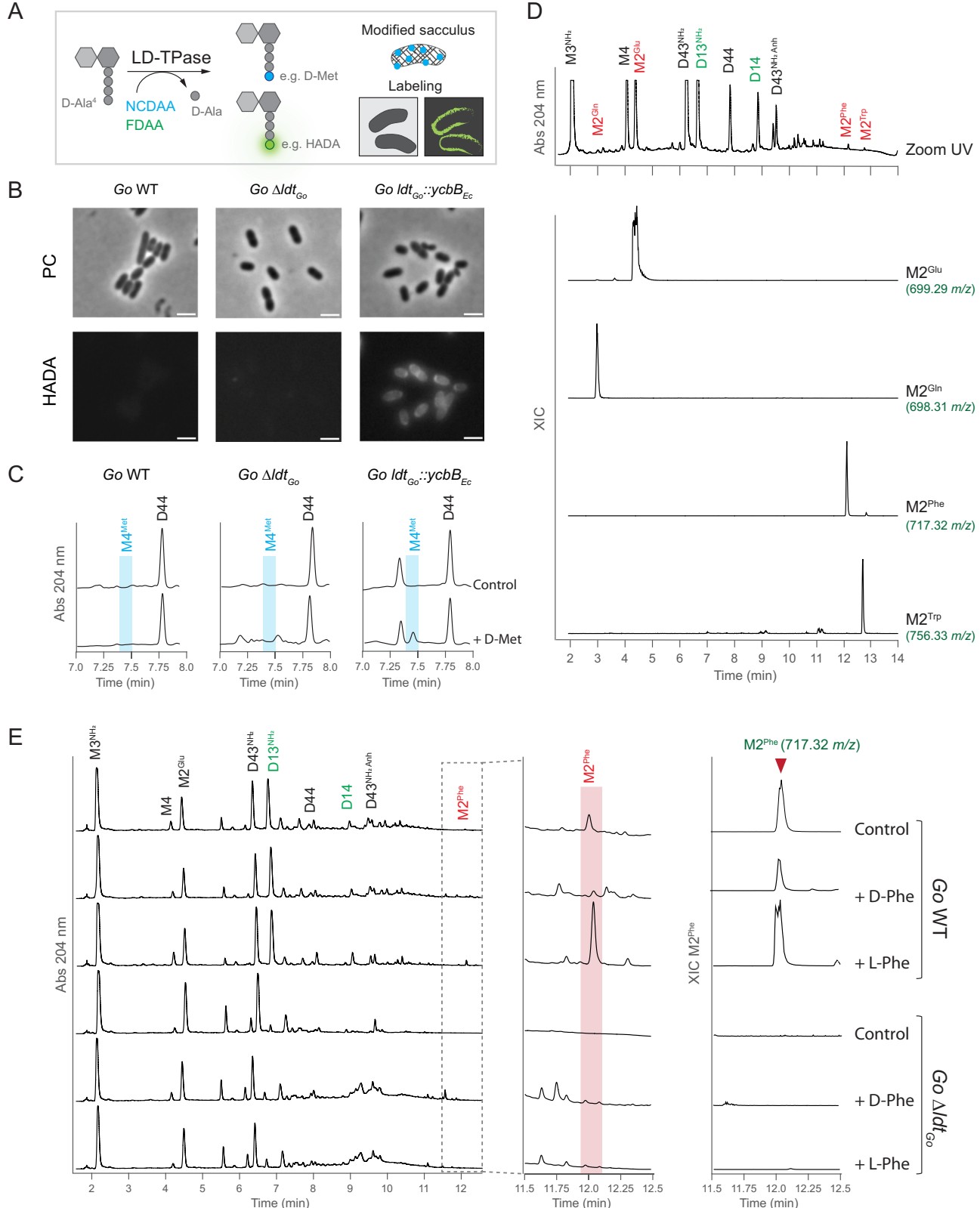

**Fig. 4 | LDT$_{Go}$ catalyzes D,L-amino acid exchange reaction. A** Scheme of the amino acid exchange reaction performed by LD3,3-TPases with non-canonical (NCDAA. e.g., D-Met) and Fluorescent (FDAA, e.g., HADA) D-amino acids. **B** Phase contrast (PC) and fluorescence microscopy of *G. oxydans* wild-type, Δ*ldt$_{Go}$* mutant and *ldt$_{Go}$::ycbB$_{Ec}$* allelic exchange cells labeled with HADA. Scale bar: 2 μm. Microscopy images shown are representative of three biological replicates. **C** Zoom-in of the UV muropeptide profiles of the same strains indicated in panel (**B**) cultured with 10 mM of D-Met or without (control). **D** Zoom-in of the UV muropeptide profile of

*G. oxydans* wild-type highlighting the dipeptide muropeptides (M2$^X$, in red) including those exhibiting amino acid exchange (M2$^{Phe}$ and M2$^{Trp}$), and the LD1,3-crosslinked muropeptides in green. The MS extracted ion chromatogram (XIC) traces are shown for the indicated M2$^X$ muropeptide species. **E** UV muropeptide profile of *G. oxydans* (*Go*) wild-type (WT) and Δ*ldt$_{Go}$* mutant strains cultured without (control) or with 10 mM of L-Phe or D-Phe (left panel). Middle panel: zoom-in of the UV muropeptide profile where the M2$^{Phe}$ muropeptide elutes. Right panel: MS extracted ion chromatogram (XIC) trace of M2$^{Phe}$.

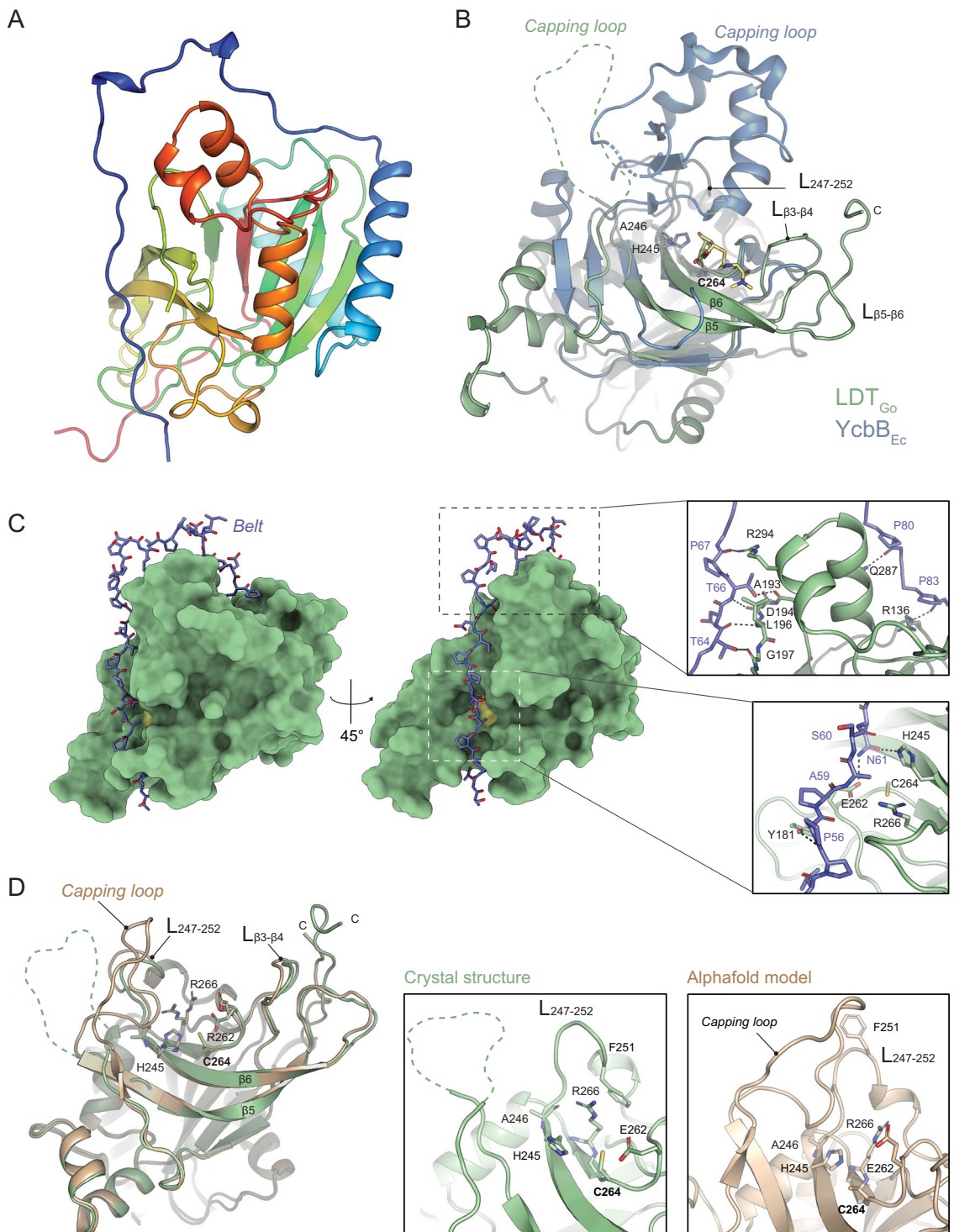

**Fig. 5 | Structure of LDT_Go. A** Overview of the LDT_Go structure, colored from the N-terminus (blue) to the C-terminus (red). **B** Superimposition of LDT_Go (green) and the catalytic domain of YcbB_Ec (blue) bound to meropenem (yellow). Three characteristic features are highlighted, i) the capping loop of both enzymes, which in LDT_Go is disordered, ii) the LDT_Go catalytic residues C264 and H245, and iii) the distinctive LDT_Go domain within the active site composed of 2 interconnected loops between the β-strands 3 and 4 (L_β3-β4) and the β-strands 5 and 6 and (L_β5-β6). **C** Surface representation of LDT_Go with the Pro-rich belt shown as sticks (C-atoms colored in purple). The catalytic Cys is shown in yellow. Zoom-in panels show the hydrogen bonds formed between the belt and the rest of the protein. **D** Superimposition of the LDT_Go structure (green) and its Alphafold2 prediction model (light brown), with the main differences highlighted in the zoom-in panels.

loop approaches the active site, the L_247–252 partially refolds and thus Phe251 moves away from the active site. This allows a reorientation of Arg266 that can then make a salt-bridge interaction with Glu262 (that in the crystal structure makes an H-bond with Asn61 from the belt) (Fig. 5).

In summary, two different conformations are observed for LDT_Go. One of these conformations involves a self-inhibited state, as observed in our crystal structure. In this state, the belt affects the conformation of catalytic residues and the loops around the active center thereby impeding access to the muropeptide substrates. The second

conformation, represented by MD simulations and the AF2 model, likely features an active state of the protein. Here, the belt has moved away, and both the capping loop and the $L_{247-252}$ region have refolded to expose the substrate-binding site.

**Structural basis of the 1,3 transpeptidation activity of LDT$_{Go}$**
We subsequently leveraged the active protein model to investigate the active site of LDT$_{Go}$ and to shed light on the mechanisms by which LD1,3- and LD3,3-TPase enzymes catalyze their specific crosslinks. As detailed before, our in vivo and in vitro experiments demonstrate that LDT$_{Go}$ produces LD1,3-crosslinks by transferring the energy of a non-terminal peptide bond. We reasoned that this unprecedented transpeptidation activity should be explained by specific structural modifications in the active site of LD1,3-TPases. We thus performed an in-depth analysis of the LDT$_{Go}$ active site compared to that of the LD3,3-TPase YcbB$_{Ec}$ in complex with meropenem (PDB ID: 6NTW) (Fig. 6A–D, Supplementary Fig. 14).

In addition to the previously mentioned reduction of the capping loop size, we observed important differences in some of the β-strands that define the active site in LDT$_{Go}$. In particular, β5 and β6 (in which the catalytic His245 lies) exhibit significantly greater length in LDT$_{Go}$ compared to YcbB$_{Ec}$. Further, these two β-strands are linked by the extended loop $L_{β5-β6}$, a structural feature that is absent in YcbB$_{Ec}$ (Fig. 6B, D, Supplementary Fig. 14).

Another distinctive feature of LDT$_{Go}$ is the presence of a long loop connecting β3 and β4, $L_{β3-β4}$, that includes a unique triad of *in-tandem* Tyr residues (Tyr180, Tyr181 and Tyr182) (Fig. 6D, Supplementary Figs. 14 and 15). The side-chain orientation of two of these Tyr residues is stabilized through the interaction with Trp231 from the $L_{β5-β6}$ (Fig. 6D). Interestingly, this aromatic triad together with the associated Trp is broadly conserved among LDT$_{Go}$ orthologs found in Acetobacteraceae species (Supplementary Fig. 15A, B). Structural predictions by AF2 of orthologues from different phyla underscore that the above-mentioned distinctive features of LDT$_{Go}$ namely, the longer β5 and β6, and loops $L_{β5-β6}$ and $L_{β3-β4}$, are structural hallmarks of this new family of LD-TPases (Supplementary Fig. 15C–I).

These modifications, together with an altered distribution of charged residues, likely account for the crosslinking specificity of LD1,3-TPases. While the acceptor groove exhibits similar basic characteristics and dimensions in both YcbB$_{Ec}$ and LDT$_{Go}$ (Fig. 6A, C), it is the donor site that stands out as notably distinct in these proteins. In LDT$_{Go}$, the donor site, marked by the position of meropenem in the YcbB$_{Ec}$:meropenem complex, is significantly shorter, measuring around 12 Å compared to the longer span of about 17 Å in YcbB$_{Ec}$. In addition, in YcbB$_{Ec}$, there is a narrow groove that extends toward the catalytic Cys residue, whereas in LDT$_{Go}$, a large and open cavity is observed near the donor site with a pronounced acidic character (Fig. 6C). MD simulations indicated that the LDT$_{Go}$ donor site can stabilize M4 to make LD1,3-crosslinked dimers. Simulations with a tetrapeptide ligand connected to a short (Fig. 6E) or longer (Supplementary Fig. 16) peptidoglycan chain indicate that the capping loop and $L_{β5-β6}$ and $L_{β3-β4}$ are crucial elements in the stabilization of glycan chains of the donor substrate. In particular, the triad of Tyr residues (Tyr180, Tyr181 and Tyr182) in loop $L_{β3-β4}$ and the Trp231 in $L_{β5-β6}$ are involved in both van der Waals and H-bond interactions with the three sugar rings (Fig. 6F). This characteristic sets LDT$_{Go}$ apart from LD3,3-TPases, which primarily target the peptide stem[30], and facilitates the approach of the NAM component of the glycan moiety of the muropeptide towards the catalytic Cys264, ultimately enabling the formation of an adduct with L-Ala[1] (Fig. 6E). Remarkably, the remaining part of the peptide moiety (the tripeptide D-Glu[2]-mDAP[3]-D-Ala[4]) is located in the extra cavity, not present in LD3,3-TPases, by the donor groove of LDT$_{Go}$. The tripeptide after the L-Ala[1] is stabilized by van der Waals interactions (mainly with Trp224 and Tyr198) and a strong polar (salt-bridge) interaction is predicted to occur between the carboxylate group of mDAP and Arg241 (Fig. 6F). The

acidic character observed in this extra region could provide a repulsive effect to avoid the carboxylate-containing residues at positions 2–4 (i.e., D-Glu-mDAP-D-Ala) being placed close to the catalytic Cys residue. Thus, our MD simulations support the preferential utilization of M3 and M4 muropeptides and explain the unique features observed in LDT$_{Go}$ in comparison with LD3,3-TPases.

**LD1,3-crosslink formation is controlled by substrate availability**
LD1,3-crosslinks are more prominent in the stationary phase peptidoglycan of *G. oxydans* than in exponential growth phase[16], suggesting LDT$_{Go}$ expression could be increased when it transitions into growth arrest. However, this is not the case as LDT$_{Go}$ expression and protein levels are comparable across growth phases (Supplementary Fig. 17A, B).

Inspired by the inverse correlation between the M4 or M3$^{NH2}$ monomer and the LD1,3-crosslinked dimer levels (Fig. 3C, Supplementary Fig. 6D), we hypothesized that generation of 1,3-crosslinks could be modulated by monomer substrate availability. Enzymes controlling the abundance of these monomers include DD-carboxypeptidases (DD-CPases) that cleave off the terminal D-Ala[5], or DD-endopeptidases (DD-EPases), which break DD-crosslinked muropeptides (e.g., D44 and D43 dimers, crosslinked between D-Ala[4]-mDAP[3]) into M4 and M3 monomers (Fig. 7A). As *G. oxydans* only encodes one putative DD-CPase (*gox0019*) and one DD-EPase with homology to PBP7 (*gox0607*), we generated individual deletion mutants and evaluated their implication on LD1,3-crosslink formation. Deletion of *gox0019* was not possible, suggesting this protein is essential in *G. oxydans*. However, the Δ*gox0607* strain was viable and exhibited a notable accumulation (ca. 50%) of DD-crosslinked peptides (Fig. 7B, Supplementary Fig. 17C, D). These findings align with its presumed function as a DD-EPase, and so we named it PBP7$_{Go}$. Remarkably, Δ*pbp7$_{Go}$* had a roughly 60% reduction in LD1,3-crosslink levels, an effect that was reverted by genetic complementation (Fig. 7C, Supplementary Fig. 17C, D), confirming the implication of this enzyme in boosting the levels of LD1,3-crosslinks in stationary phase.

To evaluate whether the reduced LD1,3-crosslink levels in the Δ*pbp7$_{Go}$* mutant is the result of a reduction of substrate flux or an effect on LDT$_{Go}$ activity (e.g., through protein-protein interactions), we generated a point mutant (S56A) in the predicted catalytic serine of PBP7$_{Go}$. Expression of the allele carrying this point mutation did not complement the lower LD1,3-crosslinking of the Δ*pbp7$_{Go}$* strain (Fig. 7B, C, Supplementary Fig. 17C). Furthermore, the Δ*ldt$_{Go}$* mutant strain presents DD-crosslink levels that are comparable to the *G. oxydans* wild-type strain, buttressing the idea that PBP7$_{Go}$ feeds tetra and tripeptide monomers to LDT$_{Go}$ for LD1,3-crosslink synthesis (Fig. 7B, Supplementary Fig. 17C). The sequential but independent action of PBP7$_{Go}$ and LDT$_{Go}$ was further confirmed by cross complementation of the Δ*pbp7$_{Go}$* strain with alternative DD-EPases such as *E. coli* PbpG$_{Ec}$ and DacB$_{Ec}$. Expression of these orthologous endopeptidases reduced DD-crosslinking by around 30% and concomitantly increased the LD1,3-crosslinking levels (Fig. 7B, C, Supplementary Fig. 17C). In sum, these results demonstrate a regulatory link between DD-crosslinking and LD-crosslinking synthesis.

Depletion of LD1,3-crosslinking has no phenotypical consequences in *G. oxydans* morphology or growth under optimal culture conditions (Fig. 7D, E, Supplementary Fig. 18A, B). However, *G. oxydans* Δ*ldt$_{Go}$* is notably more sensitive than the wild-type strain when challenged with cell wall or membrane-active antibiotic such as ampicillin (4 times lower MIC), fosfomycin and deoxycholate (Fig. 7D, E, Supplementary Fig. 18C). Interestingly, these phenotypes were alleviated by genetic complementation with LDT$_{Go}$ (Δ*ldt$_{Go}$* pLDT$_{Go}$) but not when the *ldt$_{Go}$* allele was replaced by YcbB$_{Ec}$ (*ldt$_{Go}$::ycbB$_{Ec}$*) (Supplementary Fig. 18C). In sum, these results suggest that LD1,3-crosslinks strengthen the *G. oxydans* cell wall to withstand cell envelope perturbations and that they are not functionally exchangeable with 3,3-type LD-crosslinks.

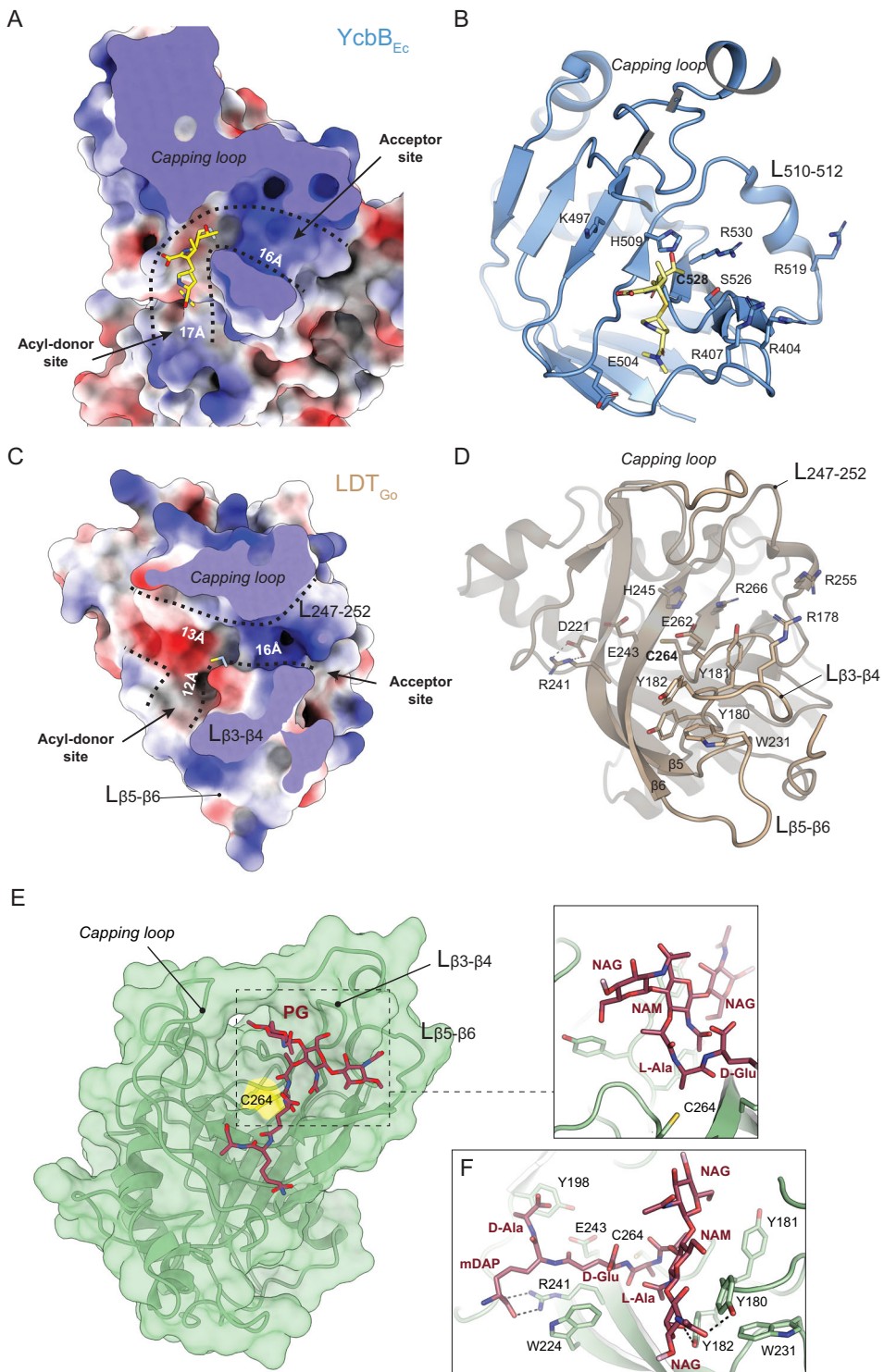

**Fig. 6 | LDT_Go active center.** Electrostatic surface representation, ranging from −13.35 to +15.88 for YcbB_Ec (**A**), and from −15.69 to +12.77 for LDT_Go (**C**) and ribbon diagram (**B**, **D**) of the donor and acceptor sites of YcbB_Ec in complex with the antibiotic meropenem (in yellow) (**A**, **B**) and of LDT_Go (**C**, **D**). The dimensions of the donor and acceptor cavities as well as the positions of the relevant loops and residues are indicated. **E** LDT_Go with the disaccharide tetrapeptide (M4) modeled in the active site. Molecular surface of LDT_Go (green) with the catalytic C264 highlighted in yellow. The muropeptide is represented as sticks (C atoms colored in dark red). The positions of the relevant loops are indicated by arrows and labeled. **F** Detailed view of the interaction of the M4 muropeptide into the LDT_Go active site. Relevant residues in the protein are represented as sticks and labeled.

## Discussion

The bacterial cell wall has been the subject of decades of research. Although some bacterial model organisms have been extensively investigated, less is known about how the cell wall is built and remodeled in other species. We found that the peptidoglycan of acetic acid bacteria was characterized for being amidated in the L-center of mDAP[3] and for presenting a previously unrecognized LD-crosslink between the L-Ala[1] and the DAP[3] [16]. Using *G. oxydans*, we report the identification, activity, and structural properties of these novel LD1,3-crosslinking enzymes.

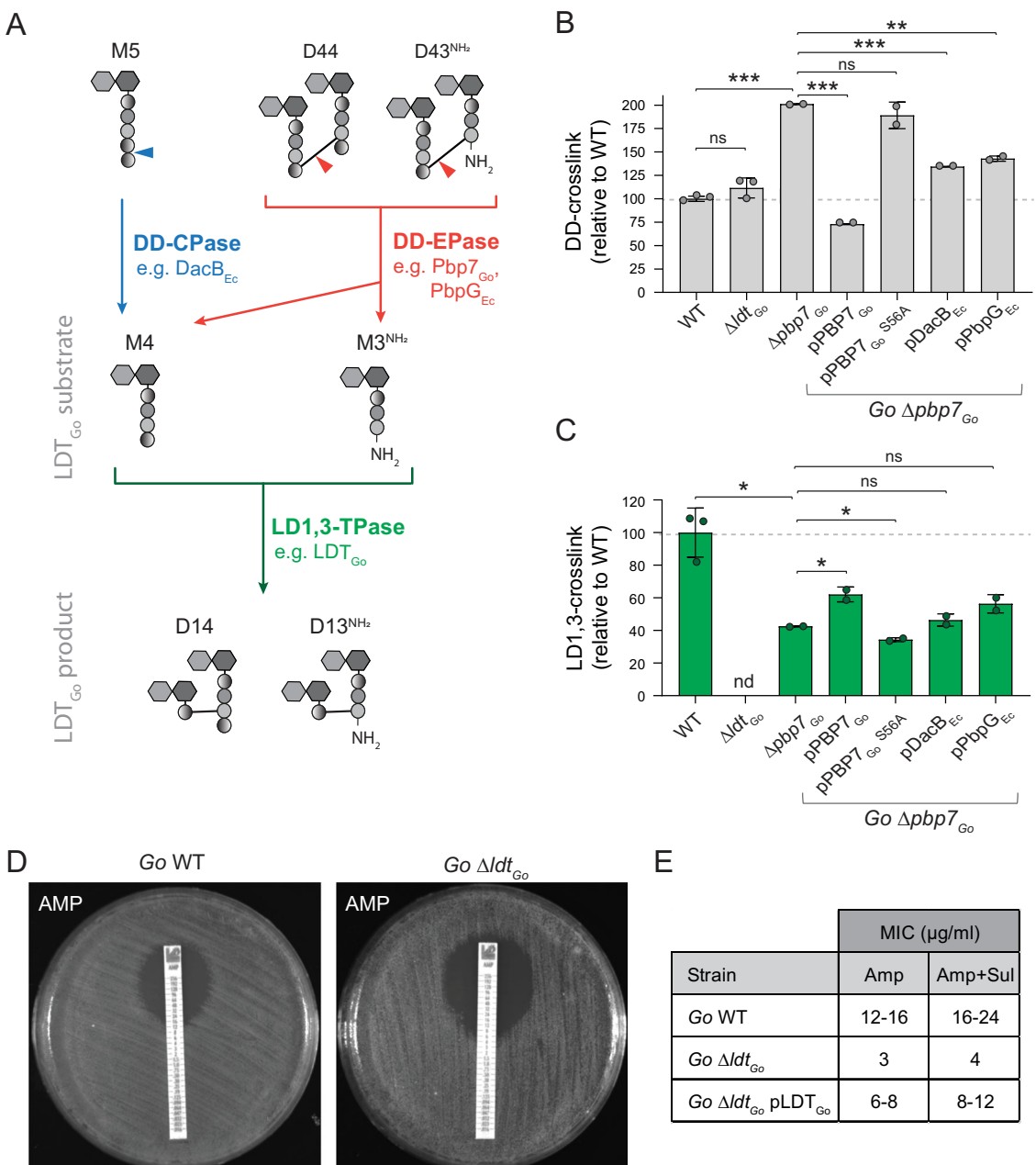

**Fig. 7 | DD-crosslinking turnover controls LD1,3-crosslinking levels. A** Scheme of the production of monomeric substrates of LDT_Go by DD-CPases (blue) and DD-endopeptidases (red). Cleavage sites are indicated by arrowheads. DD-crosslinking (**B**) and LD-crosslinking (**C**) levels for *G. oxydans* (*Go*) wild-type (WT), Δ*ldt_Go*, Δ*pbp7_Go* and the indicated complemented strains. Relative molar abundances of the DD- and LD1,3-crosslinked muropeptides were quantified and represented relative to the *Go* WT levels. **D** Representative images of *G. oxydans* WT and Δ*ldt_Go* treated with Ampicillin MIC test strips. **E** Table indicating MIC values for Ampicillin (Amp) and Ampicillin/Sulbactam (Amp+Sul) for the *G. oxydans* WT, Δ*ldt_Go* and Δ*ldt_Go* pLDT_Go complemented strain. Error bars in graphs **B**, **C** represent standard deviation from mean. Significant differences (unpaired *t*-test, two-tailed) are indicated: *$p$-value < 0.05; **$p$-value < 0.001; ***$p$-value < 0.0001; ns: not significant. Source data for **B**, **C** and exact $p$-values are provided as a Source Data file.

Although LD1,3-TPases share an active YkuD domain with the well-studied LD3,3-TPases, these enzymes exhibit poor sequence identity (e.g., LDT_Go has less than 10% of identity with YcbB_Ec) and produce chemically different crosslinks. Yet, our biochemical data suggested that these differences were likely not at the acceptor site given that like YcbB_Ec, LDT_Go can use muropeptides of varying peptide lengths such as tripeptides, tetrapeptides (e.g., Fig. 1A), and even pentapeptides (Supplementary Fig. 19). Instead, structural differences in their donor site likely explains how LD1,3-TPases make L-Ala^1-DAP^3 crosslinks and the unique nature of these enzymes in engaging non-terminal amino acids in the crosslinking process, in contrast to the rigid specificity of LD3,3-TPases and DD-TPases, which exclusively operate on tetra and pentapeptide substrates respectively.

Indeed, the donor site of the catalytic domain is very different between 1,3 and 3,3 LD-TPases. We identified a new region in our LDT_Go structure that consists of 2 interconnected loops between β-strands, which are broadly conserved among LD1,3-TPases. The structural properties of these loops shed light onto two important mechanistic questions: (i) how does the acyl-enzyme complex occur? and, (ii) how can these enzymes use different donor muropeptide substrates? The answer to the first question lies in one of the long linker loops. In LDT_Go and other Acetobacteria, this loop includes a trio of consecutive

tyrosines that likely stabilize the glycan moiety of the donor muropeptide, thereby bringing the L-Ala in position 1 of the peptide to just 3.3 Å from the catalytic Cys. In other LD1,3-TPases, this triad is often replaced by a DHF sequence; however, these residues could form the same glycan stacking interactions (Supplementary Fig. 15A, B). Interestingly, networks of aromatic residues have been previously implicated in glycan stabilization in some endolysins of phages suggesting this strategy might also be used in other cell wall acting enzymes[31].

The interaction of the catalytic Cys with the L-Ala[1] raises a dilemma: how do LD1,3-TPases accept longer muropeptides such as M4? In *Mycobacterium tuberculosis*, the Trp340 residue of the LD3,3-TPase LdtMt$_2$ has been suggested to cause steric hindrances that restrict the enzyme's substrate preference for tetrapeptides[32]. Instead, LDT$_{Go}$ has a large acidic cavity observed near the donor site that is compatible with the accommodation of muropeptide donor substrates of variable peptide length. This acidic character would prevent stabilization of the carboxylate-containing peptides at positions 2–4 (D-Glu$^2$-mDAP$^3$-D-Ala$^4$) close to the catalytic Cys264 thus favoring interaction with L-Ala[1].

The flexibility of LD1,3-TPases to accept various donor muropeptides might contribute to crosslinking homeostasis. It is known that several species increase their amount of LD3,3-crosslinks in stationary phase[9,33]. Similarly, LD1,3-crosslinking increases in *G. oxydans* and other Acetobacteria when the cultures reach high population densities[16]. However, while such an upshift is often the result of a stress-driven upregulation of the enzyme[9,32,34,35], we found that in *G. oxydans*, LDT$_{Go}$ levels are comparable across growth phases. Our results show instead that the endopeptidase PBP7$_{Go}$ plays a critical role in regulating substrate availability for LD1,3-crosslinking (Fig. 7). Linking DD-crosslink turnover with LD1,3-crosslink synthesis could help *G. oxydans* maintain peptidoglycan homeostasis, particularly during stationary phase of growth, when DD-transpeptidation declines due to reduced de novo synthesis and pentapeptide levels[36]. Interestingly, the expression of PBP7$_{Go}$ is not upregulated in stationary phase (Supplementary Fig. 17B) suggesting a more complex regulation of the synthesis of LD1,3-crosslinks. It is worth noting the possibility that certain hydrolases can also regulate the levels of this type of crosslink. However, *G. oxydans* encodes no homologs of the LD-EPase MepK from *E. coli* or other M15 peptidases[37]. Therefore, further mechanistic studies would be required to explore the role of other hydrolases in LD1,3-crosslink homeostasis.

To this end, we previously reported that the peptidoglycan of acetic acid bacteria is highly amidated in exponential phase, a modification that dramatically vanishes coinciding with the production of LD1,3-crosslinks in stationary phase[16]. As this modification can negatively affect the activity of distinct endopeptidases[16], it is possible that in addition to the role of DD-endopeptidases as substrate suppliers, mDAP amidation might play a role in controlling the use of these monomers in LD1,3-crosslinking. Identification of the genetic determinants of mDAP amidation and its growth phase dependent regulation will likely provide new insights into crosslinking homeostasis of Acetobacteria.

An additional layer of complexity in the regulation of LD1,3-crosslinks emerges from the presence of a Pro-rich belt blocking the active site in our LDT$_{Go}$ structure. This property, which was not predicted by the AF2 model, suggests a potential self-modulation of LDT$_{Go}$ activity. Our results indicate that this protein is attached to the membrane in vivo and thus it is unlikely that the belt needs to be cleaved to activate the protein. Instead, we hypothesize that a reversible binding of the belt can restrict or facilitate access of LDT$_{Go}$ to the peptidoglycan layer in addition to unlocking its active site (Supplementary Fig. 20). Along this line, it has been previously proposed that Pro-rich regions are associated with the bacterial cell wall[38,39] and N-terminal disordered extensions allow lipoproteins to cross the peptidoglycan and interact with their PBP partners[40,41]. As the Pro-rich

N-proximal belt does not seem to be restricted to Acetobacterial LDT$_{Go}$ orthologs, we hypothesize that the reversible belt-inhibition of LD1,3-TPases should respond to general stimuli like those found in stationary phase (when LD1,3-crosslinking peaks up) rather than a specific lifestyle or environmental context.

Although LD1,3-crosslinks are not essential for *G. oxydans*, they do increase cell wall integrity during diverse cell envelope stresses, including β-lactam antibiotics and membrane perturbations. Further, we previously demonstrated that this LD1,3-crosslink is immune to potential attacks caused by peptidoglycan-degrading predatory endopeptidases such as those delivered by type VI secretion systems[16]. Based on our data, we propose a hypothesis that parallels the endogenous function of PBP7$_{Go}$ as substrate supplier. We suggest that when exogenous DD-endopeptidases attack (or when PBPs are rendered inactive by β-lactams), the local availability of muropeptide monomers increases. Consequently, this would stimulate LD1,3-crosslinking activity, effectively repairing the damage by replacing the broken bond with an alternative, more resilient one within the peptidoglycan structure. As bacteria encoding LD1,3-TPases include opportunistic pathogens such as *Granulibacter bethesdensis* and *B. cenocepacia*[42,43], the activity of these enzymes might be a chemotherapeutic target to sensitize bacteria to β-lactams. Although it remains to be investigated whether the phenotypes observed for the *ldt$_{Go}$* mutant are caused solely by a reduction of the peptidoglycan crosslinking levels or if potential interactions between the peptidoglycan and the outer membrane are also compromised, the observed D,L-amino acid exchange activity of these enzymes supports this possibility. Future research will determine whether LD1,3-TPases can, like their Alphaproteobacterial LD3,3-TPase counterparts, tether the peptidoglycan to outer membrane proteins[10,44].

## Methods

### Microbiology and molecular biology

All *Gluconobacter oxydans* strains used in this study were derived from the sequenced DSM-7145[45]. Strains used in this study are listed in Supplementary Table 2. *G. oxydans* cultures were routinely grown aerobically to stationary phase in YP medium (5 g Yeast extract, 5 g peptone per liter) without or with 3% mannitol (YPM) at 30 °C. *E. coli*, *V. cholerae* and *Burkholderia* strains were grown aerobically in Luria broth (LB) at 37 °C. For agar plates, 15 g/l agar was added to the medium. Ampicillin (Amp) 100 μg/ml, Kanamycin (Kan) 50 μg/ml and Cefoxitine (Cfx) 50 μg/ml were used when required. In the amino acid exchange experiments, 10 mM D- or L-amino acids were used as supplements. Plasmids (Supplementary Table 3) were constructed by standard DNA cloning techniques. Constructs were PCR-amplified and cloned into the pSEVA238, pET28b(+), pET22b(+) and pKOS6b plasmids as indicated in the text. Mutants in *G. oxydans* were constructed as described in ref. 46 with some modifications. pKOS6b derivative plasmids carrying the 1 Kbp flanking regions of the gene of interest were introduced by conjugation. Stationary phase *G. oxydans* recipient strain and S17-1 λ-PIR donor strains were washed of antibiotics, mixed in equal ratios, and placed on a YPM agar plate in a drop. After 24 h incubation, the cells were washed from the plate with fresh YPM medium and the resulting cell suspension was plated on YPM plates containing Kanamycin 50 μg/ml and Cefoxitine 50 μg/ml, as *G. oxydans* possesses a natural resistance to this antibiotic. Ectopic complementation of *G. oxydans* mutant strains was performed using pSEVA238 derivatives listed in Supplementary Table 3, transferred by conjugation from S17-1 λ-PIR donor strains as described above. Induction of the gene expression was achieved by the addition of 3-methylbenzoic acid 1 mM. For peptidoglycan analysis of the LDT activities in vivo in *E. coli*, BL21 (DE3) cultures transformed with the corresponding pET22b(+) and pET28b(+) plasmid derivatives (Supplementary Tables 2 and 3) were grown to OD$_{600}$ 0.4 units and induced with 1 mM IPTG during 3 h.

## Growth curves, viability assays, and antibiograms

For growth curves, stationary phase cultures were normalized to an $OD_{600}$ of 0.04 and 20 µl were used for inoculating 96-well plates containing 180 µl of fresh YPM medium. At least three replicates per strain and condition were inoculated in two independent experiments. Optical density was monitored using an Eon Biotek plate reader and Biotek Gen5 [v.08] software (Biotek, Winooski, VT, USA), at 5 min intervals at 30 °C. Viability assays were done with normalized overnight cultures subjected to serial 10-fold dilution. Five-microliter drops of the dilutions were spotted onto the indicated agar plates and incubated at the appropriate temperature for 24–48 h prior to image acquisition. For antibiograms, overnight cultures of the different strains were normalized to $OD_{600}$ 0.05 and spread with a sterile cotton swap over a YPM agar plate. After the plates were air-dried, the MIC Test Strips (Liofilchem, Italy) were added. Plates were incubated for 24–48 h at 30 °C and the inhibition zone diameter was measured following the manufacturer's instructions.

## Protein expression and purification

The *G. oxydans*, *B. cenocepacia* and *E. coli* genes encoding $LDT_{Go}$, $LDT_{Go}$ C264A, $LDT_{Bcn}$, $LDT_{Bcn}$ C354A, $LdcA_{Ec}$[25], $LdtA_{Vc}$[9] were cloned on pET28b(+) (Novagen) with C-terminal His-tags for expression in *E. coli* BL21(DE3) cells[47]. Bacteria were cultured in Terrific Broth (24 g/l Yeast extract, 20 g/l tryptone, 4 ml/l glycerol, 0.017 M $KH_2PO_4$, 0.072 M $K_2HPO_4$) and expression was induced at $OD_{600}$ 0.4 with 1 mM IPTG and left overnight at 16 °C. Cell pellets were resuspended in PBS with a Complete Protease Inhibitor Cocktail Tablet (Roche) and lysed by 2 passes through a French press at 10,000 psi. After centrifugation (30 min, $100,000 \times g$), $LdcA_{Ec}$ was purified from the cleared lysates via Ni-NTA agarose columns (Qiagen) and eluted with a discontinuous imidazole gradient using an ÄktaGo system. For $LDT_{Go}$, $LDT_{Go}$ C264A, and $LDT_{Bcn}$, the pellet was treated with 0.1% (v/v) Triton X-100 overnight. The solubilized fraction was centrifuged (30 min, $100,000 \times g$) and the supernatant was purified via Ni-NTA agarose columns (Qiagen) and eluted with a discontinuous imidazole gradient using an ÄktaGo system, with buffers containing 0.1% (v/v) Triton X-100. For purification of PelB-$LDT_{Go}$, used for crystallographic studies, a C-terminal 6His-tagged version was cloned onto pET22b(+) (Novagen) for expression in *E. coli* BL21(DE3) cells using the signal peptide PelB sequence encoded in the plasmid. Cell pellets were resuspended in sodium phosphate buffer (PBS) with a Complete Protease Inhibitor Cocktail Tablet (Roche) and lysed using French press at 10,000 psi. After centrifugation (30 min, $100,000 \times g$), the cleared lysate was purified via Ni-NTA agarose columns (Qiagen) and eluted with a discontinuous imidazole gradient using an ÄktaGo system. Purified fractions were loaded on a size exclusion chromatography (SEC) Superdex 200 Increase 10/300 GL column equilibrated with 100 mM citrate/citric acid buffer pH 5 with 300 mM NaCl. Purified proteins were visualized by SDS-PAGE electrophoretic protein separation and quantified by Bio-Rad Protein Assay (Bio-Rad). The proteins were either stored at 4 °C for immediate use, or at −80 °C after the addition of 10% (v/v) glycerol.

## Sacculi and muropeptides preparation

Cells from 0.2 l cultures of overnight stationary phase ($OD_{600} = 4.0$) or 1 L exponential phase ($OD_{600} = 0.5$) were pelleted at $5250 \times g$ and resuspended in 5 ml of PBS, added to an equal volume of 10% (w/v) SDS in a boiling water bath and vigorously stirred for 4 h, then stirred overnight at room temperature. The insoluble fraction (peptidoglycan) was pelleted at $400,000 \times g$, 15 min, 30 °C (TLA-100.3 rotor; OptimaTM Max ultracentrifuge, Beckman) and resuspended in Milli-Q water. This step was repeated 4–5 times until the SDS was washed out. Next, sacculi were treated with Pronase E 0.1 mg/ml at 60 °C for 1 h and then boiled in 1% SDS for 2 h to stop the reaction. After SDS was removed as described previously, sacculi samples were resuspended in

200 µL $H_2O$ and used as substrate in in vitro reactions, or in 200 µL of 50 mM sodium phosphate buffer pH 4.9 for subsequent digestion with mutanolysin. For preparation of muropeptides, samples were digested overnight with 30 µg/ml mutanolysin (from *Streptomyces albus*) at 37 °C. Mutanolysin digestion was stopped by heat-inactivation (boiled for 5 min). Coagulated protein was removed by centrifugation ($20,000 \times g$, 15 min). The supernatants (soluble muropeptides) were subjected to sample reduction. First, pH was adjusted to 8.5–9 by the addition of borate buffer 0.5 M pH 9 and then N-acetylmuramic acid residues were reduced to muramitol by sodium borohydride treatment ($NaBH_4$ 10 mg/ml final concentration) during 30 min at room temperature. Finally, pH was adjusted to 2.0–4.0 with orthophosphoric acid 25% (v/v) prior to analysis by LC.

## Peptidoglycan analysis

Chromatographic analyses of muropeptides were performed by Ultra Performance Liquid Chromatography (UPLC) using Empower 3.6 software (Waters) on an UPLC system (Waters) equipped with a trapping cartridge precolumn (SecurityGuard ULTRA Cartridge UHPLC C18 2.1 mm, Phenomenex) and an analytical column (BEH C18 column (130 Å, 1.7 µm, 2.1 mm by 150 mm; Waters, USA). Muropeptides were detected by measuring the absorbance at 204 nm using an ACQUITY UPLC UV−visible Detector. Muropeptides were separated using a linear gradient from buffer A (Water + 0.1% (v/v) formic acid) to buffer B (Acetonitrile 100% (v/v) + 0.1% (v/v) formic acid) over 15 min with a flowrate of 0.25 ml/min. Individual muropeptides were quantified from their integrated areas using samples of known concentration as standards. Identity of the muropeptides was confirmed by MS and MS/MS analysis, using a Xevo G2-XS Q-tof system (Waters Corporation, USA). The instrument was operated in positive ionization mode. Detection of muropeptides was performed by $MS^E$ (method of data acquisition that records exact-mass data for every detectable component and its fragment ions) to allow for the acquisition of precursor and product ion data simultaneously, using the following parameters: capillary voltage at 3.0 kV, source temperature to 120 °C, desolvation temperature to 350 °C, sample cone voltage to 40 V, cone gas flow 100 l/h, desolvation gas flow 500 l/h and collision energy (CE): low CE: 6 eV and high CE ramp: 15–40 eV. Mass spectra were acquired at a speed of 0.25 s/scan. The scan was in a range of 100–2000 $m/z$. Data acquisition and processing was performed using UNIFI 1.8.1 software (Waters Corp.). Chromatograms shown are representative of three biological replicates. The quantification of muropeptides was based on their relative abundances (relative area of the corresponding peak) and relative molar abundances, as indicated elsewhere[48]. The relative molar abundance is the percentage of the peak area of a muropeptide, divided by its molecular weight, compared to the sum of peak areas in the chromatogram. A table of all the identified muropeptides and the observed ions is provided (Supplementary Table 5).

## In vitro activity assays

The $LDT_{Go}$, $LDT_{Bcn}$, $LDT_{Bcn}$ C354A, $LdtA_{Vc}$, $LdcA_{Ec}$ and KP27 reactions were performed in 50 µl reactions using 0.1 mg/ml of purified enzymes with 1 mg/ml of sacculi isolated as described above from stationary phase cultures from *G. oxydans*, *G. oxydans* $\Delta ldt_{Go}$, *V. cholerae* WT or $\Delta dacA1$ mutant. To generate M3 and D-Met labeled sacculi, $LdcA_{Ec}$ or $LdtA_{Vc}$ with 10 mM of D-methionine (D-Met), respectively, were incubated for 2 h at 37 °C prior to heat-inactivation (boiled for 5 min) and addition of $LDT_{Bcn}$. LD1,3-TPase reactions were carried out in LD buffer (50 mM Tris HCl, pH 8, 50 mM NaCl) overnight at 30 °C. For control reactions with KP27, KP buffer was used (20 mM Tris-HCl, pH 8, 1 mM $MgCl_2$, 1 mM $ZnCl_2$) at 37 °C for 90 min. Reactions were heat-inactivated (boiled for 5 min), and fractions were separated by centrifugation at $20,000 \times g$ for 15 min. For analysis of the insoluble product (pellet), reactions were finally treated with 50 µg/ml mutanolysin for 2 h at 37 °C, heat-inactivated (boiled for 5 min) and, followed by

**Table 1 | Data collection and refinement statistics**

|  | LDT$_{Go}$ |
|---|---|
| Wavelength (Å) | 0.8731 |
| Resolution range (Å) | 48.41 – 1.732 (1.794 – 1.732) |
| Space group | P 21 21 21 |
| Unit cell |  |
| a, b, c (Å) | 52.78, 56.69, 93.02 |
| α, β, γ (°) | 90, 90, 90 |
| Total No. of reflections | 163,948 (7995) |
| Unique reflections | 54,354 (4155) |
| Multiplicity | 3.0 (1.9) |
| Completeness (%) | 98.48 (87.39) |
| Mean I/σ(I) | 6.68 (0.74) |
| Wilson B-factor (Å$^2$) | 24.55 |
| R-merge | 0.1004 (0.9055) |
| R-meas | 0.1206 (1.16) |
| R-pim | 0.06601 (0.7139) |
| CC1/2 | 0.995 (0.355) |
| CC* | 0.999 (0.724) |
| Reflections used in refinement | 29,296 (2335) |
| Reflections used for R-free | 1530 (119) |
| R-work | 0.1706 (0.3252) |
| R-free | 0.2051 (0.3319) |
| CC1/2 | 0.995 (0.355) |
| CC* | 0.999 (0.724) |
| No. of non-hydrogen atoms | 2191 |
| macromolecules | 2056 |
| Ligands | 2 |
| Solvent | 133 |
| Protein residues | 263 |
| RMS (bonds) (Å) | 0.006 |
| RMS (angles) (°) | 0.72 |
| Ramachandran favored (%) | 96.14 |
| Ramachandran allowed (%) | 3.86 |
| Ramachandran outliers (%) | 0.00 |
| Rotamer outliers (%) | 0.00 |
| Clashscore | 0.98 |
| Average B-factor (Å$^2$) | 27.10 |
| Macromolecules | 26.71 |
| Ligands | 34.35 |
| Solvent | 32.91 |

Statistics for the highest-resolution shell are shown in parentheses.

centrifugation at 20,000 × g for 15 min to remove precipitated proteins. Samples were subjected to reduction, pH adjusted, and analyzed by LC as described above. Analysis of the soluble fraction of the LD1,3-TPase in vitro reactions was performed for identification of released tripeptides. Non-reduced samples were pH adjusted and analyzed by LC-MS, as described above. To determine LDT$_{Bcn}$ inhibition, Ampicillin (100 μg/ml), Imipenem (100 μg/ml) and Copper (1 mM) were added to the reaction before the addition of the enzyme.

### Crystallization and structure determination
Purified LDT$_{Go}$ was concentrated on a 10 kDa cutoff Amicon Ultra Centrifugal Filter (Merck-Millipore) to ~10 mg/ml and loaded on a Superdex-200 increase 10/300 GL column equilibrated in 100 mM citrate/citric acid buffer pH 5.0, 300 mM NaCl. Protein peak fractions were concentrated further to 19 mg/ml. Crystals were grown at 20 °C

by sitting drop vapor diffusion, using a 1:1 protein to reservoir ratio, in the A1 condition from the Morpheus screen, which contains 30 mM Magnesium chloride hexahydrate; 30 mM Calcium chloride dihydrate, 0.1 M Imidazole; MES monohydrate (acid) pH 6.5 and 20% v/v PEG 500* MME; 10% w/v PEG 20000. Crystals first appeared after ~3 days, were fished after 3 weeks and flash-frozen in liquid nitrogen. X-ray diffraction data was collected on the ID23-2 beamline at the European Synchrotron Radiation Facility, Grenoble, France[49]. The data was processed using XDS[50]. The crystal belonged to P2$_1$2$_1$2$_1$ space group and contained a single molecule in the asymmetric unit. The crystallographic phase problem was solved by using the AlphaFold2 model, generated in ColabFold [v1.0][51] using standard settings. This model was processed in Phenix [v1.21][52] and the first 53 residues were removed to have a suitable model for molecular replacement, which was done with Phaser [2.8.3][53]. Coot [0.9.5][54] was used to build the model and the structure was refined using Refmac5 [v5.8.0267][55] and Phenix refine [v1.13][52]. For complete data collection and refinement statistics, see Table 1. The final model was validated using MolProbity[56]. Atomic coordinates and structure factors of the LDT$_{Go}$ structure have been deposited in the Protein Data Bank (PDB ID: 8QZG).

The newer AF2 LDT$_{Go}$ and homolog proteins models were made in ColabFold [v1.5.2][51], with standard settings except that the number of cycles was increased to 10 and with the use of templates and amber relaxation. The generated AF2 models are available as PDB files in Supplementary Data 1.

### Molecular dynamics (MD) simulations
Both the full-length AF2 model and the gap-filled X-ray crystal structure of LDT$_{Go}$ were immersed in a rectangular box of TIP3P waters, neutralized with counterions, energy minimized, and subjected to MD simulation under periodic boundary conditions for up to 400 ns using the *pmemd.cuda* module of AMBER 22 (https://ambermd.org/), following a previously described protocol[57] that included a heating phase lasting 40 ps and an equilibration phase lasting 100 ps. The Berendsen thermostat and barostat were used in all cases. The peptidoglycan (PG) strand was modeled using the structure reported by[58] as "template2" and manually docked into LDT$_{Go}$ (N-capped residues 72-336) to provide a complex that was simulated under identical conditions (Supplementary Table 6). AMBER ff19SB and GLYCAM_06j parameters were assigned to peptide and glycan atoms, respectively. Electrostatic interactions were represented using the smooth particle mesh Ewald method[59] with a grid spacing of 1 Å and the cutoff distance for the non-bonded interactions was 9 Å. Positional restraints (2 kcal mol$^{-1}$ Å$^{-2}$) on C1 and C4 atoms of the alternating 1–4 linked N-acetylglucosamine and N-acetylmuramic acid (AMU) subunits were employed to preserve the straightness of the peptidoglycan's glycan strand. To ensure the correct orientation of the tetrapeptide attached to the central AMU subunit, a biasing harmonic potential of 10 kcal mol$^{-1}$ Å$^{-2}$ was included in the force field—as an extra term with the form of a flat well with parabolic sides—to keep the Cys264(SG)–L-Ala(C) and Arg241(N)–L-Ala(O) distances within a range compatible with nucleophilic attack (3.9–4.0 Å) and hydrogen bond formation (2.9–3.1 Å), respectively. The collected coordinates were analyzed with the aid of the *cpptraj* module in AMBER. The initial coordinate and simulation input and output PDB files are available in Supplementary Data 1.

### Detection of LDT$_{Go}$ by Western blot
For localization of LDT$_{Go}$ in the different cellular compartments, *G. oxydans ldt$_{Go}$::ldt$_{Go}$6his* was grown in YPM medium to stationary phase (OD$_{600}$ 3.0 units). The samples were lysed using a French press at 10,000 psi. The membrane and soluble fractions were separated by ultracentrifugation at 75,000 × g for 60 min. The membrane fractions were resuspended in 200 μl PBS and the soluble fraction was concentrated to a similar level using Amicon$^R$ Ultra 15 centricons (Millipore). 15 μg of membrane and soluble fraction were loaded after

normalization by total protein quantification using the Bio-Rad Protein Assay (Bio-Rad) onto a 12% acrylamide gel. For the quantification of $LDT_{Go}$-His in the different growth phases, cultures of *G. oxydans* $ldt_{Go}::ldt_{Go}6his$ were collected at $OD_{600}$ 0.3 (Exp), 1.2 (Early Sta) and 3 (Late Sta). The samples were OD normalized and prepared as described above. For quantification, 15 µg of the membrane fraction was loaded onto the 12% acrylamide gel.

After transfer of proteins to a PVDF membrane (Immobilon-PVDF membrane, Millipore, ref: IPVH00010), Western blotting was performed using specific anti-His antibody as primary antibody (mouse anti-$(H)_5$ antibody, Qiagen, ref: 34660, dilution 1:10,000) and anti-mouse antibody as secondary antibody (rabbit anti-mouse IgG – HRP-conjugated, Sigma, ref: A9044-2ML, dilution 1:30,000). His-tagged proteins were detected by addition of SuperSignal West Pico PLUS Chemiluminiscence substrate (Thermo Scientific, ref: 34577) and Fuji LAS-3000 Imaging System was used for image acquisition.

### Beta-galactosidase assay

The presumed promoter region corresponding to the 500 bp upstream from the putative *gox1073-1074* operon, was cloned into the promoter probe plasmid pSEVA235, functional in *G. oxydans*. Beta-galactosidase activity of the promoter-lacZ translational fusion was measured through *o*-nitrophenyl-β-d-galactopyranoside (ONPG) cleavage by the product of the *lacZ* reporter gene ($OD_{420}$ and $OD_{550}$ were acquired using an Eon Biotek plate reader and Biotek Gen5 [v.08] software (Biotek, Winooski, VT, USA)), and specific activity was calculated in Miller units[59]. Cultures of *G. oxydans* wild-type carrying pSEVA235-$Pldt_{Go}$ derivative (and pSEVA235 empty plasmid as negative control) were grown in YPM medium at 30 °C to $OD_{600}$ 0.3 units (Exp), 1.2 units (Early Sta) or 3 units (Late Sta) phase. Aliquots (100 µL) of three different subcultures were collected, and cells were permeabilized and assayed in triplicate for each strain as previously described[60].

### Quantitative real-time PCR (qRT-PCR)

The expression levels of $ldt_{Go}$ and $pbpb7_{Go}$ genes in exponential and stationary phase was analyzed by RT-PCR. RNA was isolated from *G. oxydans* wild-type cultures at $OD_{600}$ 0.3 units (Exp) or 3 units (Sta) using the RNeasy Mini Kit (Qiagen), following the manufacturer's instructions. Reverse transcription was performed with the High-Capacity cDNA Reverse Transcription Kit (Thermo Fisher), using aliquots of 5 µg total RNA. The synthesized cDNA was purified using a QIAquick PCR purification kit (Qiagen), and its concentration was determined spectrophotometrically in a Nano-drop Lite spectrophotometer (Thermo Scientific). Real time quantitative PCR (qRT-PCR) was performed using an $iQ^{TM}5$ Multicolor Real-Time PCR Detection System (BIO-RAD) with qPCRBIO SyGreen Mix fluorescein (PCR BIOSYSTEMS). Master mixes were prepared as recommended by the manufacturer, with qRT-PCR primers listed in Supplementary Table 4. Two independent experiments were carried out in triplicates for each data point. The relative quantification in gene expression was determined using the $2^{-\Delta\Delta Ct}$ method[61], using $recA_{Go}$ *(gox1522)* gene as control.

### Microscopy

For imaging, bacteria were immobilized on YPM pads containing 1% agarose. Phase-contrast microscopy was performed using a Zeiss Axio Imager.Z2 microscope (Zeiss, Oberkochen, Germany) equipped with a Plan-Apochromat × 63 phase-contrast objective lens and an ORCA-Flash 4.0 LT digital CMOS camera (Hamamatsu Photonics, Shizuoka, Japan), using the Zeiss Zen 2 Blue edition [v2.0.0.0] software. Image analysis and processing were performed using Fiji/ImageJ [v1.53][62] and the MicrobeJ plugins[63]. Microscopy images shown are representative of three biological replicates. For incorporation of FDAA, 1 mM of

HADA[26] was added to each culture at exponential phase and incubated for 30 min. Cultures were then quenched with cold methanol in ice for 20 min and visualized as described.

### Bioinformatic analyses

Orthologue sequences of $LDT_{Go}$, corresponding to LD1,3-TPases, were searched by BLAST using the NCBI non-redundant protein database. Results were filtered using the following criteria: E-value > 10, coverage > 0.5, identity > 30. LD3,3-TPases (YcbB-like orthologues) were downloaded from OrthoDBv11 (LD-transpeptidase group 29809748at2)[64]. For construction of a phylogenetic tree and conservation of proteins, we produced a tree of all species on OrthoDBv11 using PhyloT v2 and orthologues of LD1,3- or LD3,3-TPases mapped against this. The final tree was visualized using iTOL v6[65]. Multisequence alignments were performed with Clustal Omega[66] or T-COFFE Expresso[67]. Alignments were visualized and analyzed using Jalview v2[68]. We used ESPript[69] for rendering sequence similarities and secondary structure information from aligned sequences, using the $LDT_{Go}$ crystal structure (PDB ID: 8QZG) as reference. Signal peptide predictions were performed with SignalP 6.0[17]. Sequence logos were generated in R v4.3 using the ggseqlogo package[70].

### Statistical analysis

Graphpad Prism 9.0 was used for graphing and analyzing most data. Statistical significance was assessed using the Student's *t*-test (unpaired, two-tailed). A *p*-value of less than 0.05 was considered statistically significant. Assays were performed with three biological replicates unless otherwise indicated.

### Reporting summary

Further information on research design is available in the Nature Portfolio Reporting Summary linked to this article.

## Data availability

The crystal structure data used in this study are available in the PDB database under accession code 8QZG. The AlphaFold2 models generated in this study are provided in Supplementary Data 1. Source data are provided with this paper.

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

## Acknowledgements

We thank the Cava lab members for insightful discussions. We are grateful to Michael VanNieuwenhze for providing HADA. We also thank David Kostner for generously providing the plasmid pKOS6b. Research in the Cava lab was supported by the Swedish Research Council (2018-02823 and 2018-05882), Umeå University, the Knut and Alice Wallenberg Foundation, and the Kempe Foundation (SMK2062). We acknowledge MAX IV Laboratory for time on Beamline BioMax under Proposal 20190808. Research conducted at MAX IV, a Swedish national user facility, is supported by the Swedish Research Council under contract 2018-07152, the Swedish Governmental Agency for Innovation Systems under contract 2018-04969, and Formas under contract 2019-02496. We acknowledge the European Synchrotron Radiation Facility for the provision of synchrotron radiation facilities (beamline ID23-2). Research in the Berntsson lab was supported by grants from the Swedish Research Council (2016-03599), the Knut and Alice Wallenberg Foundation, and the Kempe Foundation (SMK-1762 & SMK-1869). Research in the Gago lab is funded by the Spanish MICINN Projects PID2019-104070RB-C22 and AEI/10.13039/501100011033.

## Author contributions

A.E. and F.C. conceived and designed the study. F.C. directed the study. A.E., L.A., and G.T. performed most of the experiments. J. tB. solved the structure. O.I. performed protein purification. V.M., J. tB. and J.A.H. provided comparative analysis and functional interpretation of the structural data. F.G. generated molecular dynamic simulations. All authors analyzed and contributed to the interpretation of the data. J.A.H., R.P.-A.B., and F.C. supervised the work and provided the funding. F.C. wrote the manuscript with support from all authors.

## Funding

## Competing interests

The authors declare no competing interests.
