## [Peer Review File · Nature Communications]

A distinctive family of L,D-transpeptidases catalyzing L-Ala-mDAP crosslinks in Alpha- and BetaproteobacteriaEditorial Note: Parts of this Peer Review File have been redacted as indicated to maintain the confidentiality of unpublished data.

Reviewer #1 (Remarks to the Author):

The manuscript by Espaillet et al. presents an in-depth characterization of an L,D-transpeptidase involved in unconventional 1,3 peptidoglycan crosslinks. The existence of such an enzyme was foreseen in a previous study by the same authors that revealed (1-3) L-Ala-mDAP cross-links in the peptidoglycan of several Proteobacteria. A very recent study (J. Biol. Chem., in press, doi.org/10.1016/j.jbc.2023.105494) identified a similar L,D transpeptidase and described a first preliminary characterization. The present study by Espaillet et al. provides a comprehensive study of this unconventional LD transpeptidase with additional and meaningful data, relative to its substrates, its 3D structure and its role in maintaining cell wall integrity. A thorough investigation of the enzymatic activity is provided using purified recombinant enzyme and purified sacculi as substrates. They provide convincing results showing that LD1,3-TPases use mucopeptides with tri or tetrapeptide chains as acyl donors, thus engaging non-terminal amino acids in the cross-linking process. This property makes these enzymes unique among characterized L,D-transpeptidases. They also obtained crystallographic structural data showing distinctive features of LD1,3-transpeptidases compared to LD3,3-transpeptidases, allowing to support their original substrate specificity. They further identified a D,D-endopeptidase (of the PBP family) involved in regulating substrate availability. Furthermore, they provide first insights into the role of the 1,3 LD crosslinks to counteract cell envelope stresses. The presented experimental data are sound and the manuscript is well written. Therefore, this study should be of great interest to the readers of Nature Communications.

I have a few specific comments:

L181-182: "without a corresponding impact on M2". Since M2 is present in very low amount in *V. cholerae* sacculi (as indicated L177), it seems difficult to conclude on the role of M2 as a donor substrate with this experiment. Please modify the text.

Fig. 3A: M2 is not visible on the chromatograms. Was it detected only by LC-MS? Indicate its position on the chromatograms in Fig. 2A.

Fig. 3B (legend) : how the "molar abundance" of each mucopeptide form was quantified?

L180-186: the data presented to conclude that M4 is a substrate of LDTBcn are quite convincing. However, the clarity of this paragraph could be improved to better explain the double activity of LDTBcn (LD-TPase and LD-endopeptidase). The decrease in M4 may be associated with either TPase or EPase activity and this is the same for D44 decrease. It may be mentioned that like for classical L,D-transpeptidase, this is in agreement with the formation of an acyl-enzyme intermediate (covalent intermediate between the catalytic Cys residue and disaccharide-L-Ala (M1)) requiring cleavage of the L-Ala1-Glu2 bond)). At least in vitro, this acyl intermediate could be further hydrolyzed leading to LD-EPase activity or reacts with an acyl acceptor M4 or D44 leading to LD-TPase activity.

L210-211: the authors indicate that they tested LDTGo activity on muramidase-digested peptidoglycan or purified mucopeptides without success. Which experiments were performed? Was the activity tested with non-reduced mucopeptides? Reduction of mucopeptides with sodium borohydride could inhibit enzymatic activity.

Fig. S9: Incorporation of Gln in M2 by exchange with D-Glu seems to be independent of LDTGo, since it is present in the deletion mutant. Consequently, the extraction ion chromatogram shown for M2Gln shown in Fig. 4D could be removed because it does not support the D,L-aminoacid exchange activity of LDTGo.

Fig. S8: Similar suggestion for the MS spectra of M2Gln shown in Fig. S8.

Minor comments:

L173: change "...upon the expression of LDTBcn " to "...upon the expression of LDTBcn in Δ ldtGo mutant (Fig. 1D)".

L194: change to "the L,D-TPase LdtA"

L204: change to " by an LD-carboxypeptidase (LdcA)"

L536: indicate the OD of overnight cultures.

L549: change to .. "N-acetyl-muramic acid residues were reduced to muramicitol".

L563: does "MS-MS/MS" mean "MS and MS/MS"?

L565: define MSE

L578: how were the sacculi prepared?

L581, 584, 586: precise "heat-inactivation"

L586: change "muramidase" to mutanolysin.

Reviewer #2 (Remarks to the Author):

Traditionally, the peptidoglycan (PG) studies were done in model bacteria such as *E. coli*, *Vibrio cholerae* or *Bacillus subtilis* and these yielded a wealth of information on the basic fundamental processes involved in bacterial cell wall biogenesis. PG is a polymer consisting of linear glycan strands attached to a short-stem peptides. In Gram-negative bacteria, the predominant components of the stem peptide include L-alanine (L-Ala1), D-glutamate (D-Glu2), meso-diaminopimelic acid (mDAP3), D-alanine (D-Ala4), and D-Ala5, with cross-links occurring between D-Ala4 and mDAP3 (4-3) or between two mDAP3 residues (3-3). While 4-3 cross-links are made by penicillin-binding proteins, 3-3 cross-links are the products of L,D-Transpeptidases.

Recent studies on the PG of non-model culturable bacteria have uncovered novel modifications in the cell wall, providing intriguing insights into their peptidoglycan biology. In the present study, Espaillet et al. describe how the non-canonical 1-3 cross-links in *Glucanobacter oxydans*, a member of alpha-proteobacteria are synthesized and regulated. Through extensive bioinformatics and biochemical approaches, the authors discovered and characterized a new class of enzymes involved in the formation of these 1,3 cross-links, providing evidence of how it differs from its similar counterparts, LDTs (3-3). Additionally, they elucidated the protein structure and offered mechanistic insights into its catalytic activity. This study is truly fascinating presenting clear data and a complete picture. Manuscript is written well with good flow and logic. I thoroughly enjoyed reading the manuscript. This study provides a major breakthrough in the field of peptidoglycan biology and will be of interest to a large class of bacterial research community.

I have no major concerns, but authors should consider the following:

1. Fig. 1A. Representation of T144 needs to be rechecked. Will the central mDAP make 4 bonds which are theoretically impossible? I think the D-ala of central monomer may make a crosslink with mDAP of right monomer.
2. Recommend adding a panel to Fig. 1 illustrating all the structures/ denotations of muropeptides including M1, M2 and others (which are already here).
3. Fig. 3A. A representation of M2 in the chromatogram is needed for the reader to compare the chromatogram for level of M2. Also label the M2 peak in the chromatogram.
4. In the results section 6 (322–324), it is suggested that three amino acids, Tyr 180, 181, and 182, play crucial role in conferring the characteristic 1-3 cross-linking properties to these enzymes. Have the authors attempted to mutate these residues and assess whether the enzyme still functions as a 1-3 cross-linker?
5. In Fig. 1, I would suggest adding a western blot showing the expression of catalytic site mutants.
6. The figures need to be enlarged and font sizes increased for ease in reading.
7. Discussion, lines 420, this paragraph discusses the regulation and homeostasis of different crosslinks. It is possible that 1,3- crosslink hydrolase may exist in these organisms and it may control the level of 1,3 crosslinks. Only the synthases need not be controlled; hydrolases may also contribute to the crosslinkage frequency.
8. May be a sentence on how these crosslinked muropeptides are recycled would help in the Discussion.
9. Line 54- comma to be included after pentapeptide.
10. Line 87- 'type' after novel can be removed.

Reviewer #3 (Remarks to the Author):

This is an interesting mechanistic study regarding a novel L,D transpeptidase that catalyzes 1-3 stem peptide cross-links. Authors employ extensive bacterial genetics techniques, paired with structural biology and molecular dynamics, to extensively characterize not only the novel transpeptidase itself but also the biochemical and mechanistic environment that lead to its activation. The text is elegantly written and the figures are quite clear. There are a few points that merit clarification, highlighted below.

Line 20, and elsewhere: authors should include a few sentences to describe the generalized term 'acetic acid bacteria', i.e., specific growth conditions, unusual environments, or metabolism.

Lines 116-117: what were the consequences of the 1-3 crosslinks on the overall bacterial phenotype? Did authors observe modifications in growth, shape, or survival capacity? It could be of interest to mention this here.

Lines 142-145: can authors mention a few prototypical strains in these phyla, and are there specific characteristics (such as typical growth environment, pathogenicity potential, etc) that could explain the need for such a transpeptidase in these strains?

Line 179: kindly clarify the need for Cu⁺² in this experiment.

Lines 212-213: can authors comment on the preference of LDT-Bcn for PG chains rather than individual muropeptides?

Lines 246-247: it would be of interest to the reader to find out at this point a few more details regarding how the crystal structure was solved. This reviewer had to look for this info in the Materials & Methods section. The structure was solved by using an AlphaFold model, but no details of the experiment were provided, neither in the Results section nor in M&M itself. Can authors comment on differences between the initial model and the final structure, especially in what relates to the belt region and other loops? This is shown in Fig 5 but not explored in the text.

Minor comments

-line 289, should read 'we subsequently leveraged ...'

- line 607, should read 'Coot was used ...'

-there are a few units missing in the crystallography table, notably in what relates to wavelength, resolution, B-factors, RMS ... in addition, the number of digits to the right of the decimal point could be homogenized.

Response to reviewers

Reviewer #1 (Remarks to the Author):

The manuscript by Espaillet et al. presents an in-depth characterization of an L,D-transpeptidase involved in unconventional 1,3 peptidoglycan crosslinks. The existence of such an enzyme was foreseen in a previous study by the same authors that revealed (1-3) L-Ala-mDAP cross-links in the peptidoglycan of several Proteobacteria. A very recent study (J. Biol. Chem., in press, doi.org/10.1016/j.jbc.2023.105494) identified a similar L,D transpeptidase and described a first preliminary characterization. The present study by Espaillet et al. provides a comprehensive study of this unconventional LD transpeptidase with additional and meaningful data, relative to its substrates, its 3D structure and its role in maintaining cell wall integrity. A thorough investigation of the enzymatic activity is provided using purified recombinant enzyme and purified sacculi as substrates. They provide convincing results showing that LD1,3-TPases use mucopeptides with tri or tetrapeptide chains as acyl donors, thus engaging non-terminal amino acids in the cross-linking process. This property makes these enzymes unique among characterized L,D-transpeptidases. They also obtained crystallographic structural data showing distinctive features of LD1,3-transpeptidases compared to LD3,3-transpeptidases, allowing to support their original substrate specificity. They further identified a D,D-endopeptidase (of the PBP family) involved in regulating substrate availability. Furthermore, they provide first insights into the role of the 1,3 LD crosslinks to counteract cell envelope stresses. The presented experimental data are sound and the manuscript is well written. Therefore, this study should be of great interest to the readers of Nature Communications.

We thank the reviewer for their detailed assessment of our manuscript and their enthusiastic comments.

I have a few specific comments:

L181-182: “without a corresponding impact on M2”. Since M2 is present in very low amount in *V. cholerae* sacculi (as indicated L177), it seems difficult to conclude on the role of M2 as a donor substrate with this experiment. Please modify the text.

We agree with the reviewer’s concern, and we have removed the sentence (line 181).

Fig. 3A: M2 is not visible on the chromatograms. Was it detected only by LC-MS? Indicate its position on the chromatograms in Fig. 2A.

The M2 was detected only by LC-MS analysis in *Vibrio cholerae* samples. Following the reviewer's suggestion, we have indicated its position in Fig. 3A.

Fig. 3B (legend) : how the “molar abundance” of each mucopeptide form was quantified?

The relative molar abundance is the percentage of the peak area of a mucopeptide, divided by its molecular weight, compared to the sum of peak areas in the chromatogram. We have clarified this in the figure legend (lines 1011-1013) and included it in the Methods section (lines 596-598).

L180-186: the data presented to conclude that M4 is a substrate of LDT_{Bcn} are quite convincing. However, the clarity of this paragraph could be improved to better explain the double activity of LDT_{Bcn} (LD-TPase and LD-endopeptidase). The decrease in M4 may be associated with either TPase or EPase activity and this is the same for D44 decrease. It may be mentioned that like for classical L,D-transpeptidase, this is in agreement with the formation of an acyl-enzyme intermediate (covalent intermediate between the catalytic Cys residue and disaccharide-L-Ala (M1)) requiring cleavage of the L-Ala1-Glu2 bond)). At least *in vitro*, this acyl intermediate could be further hydrolyzed leading to LD-EPase activity or reacts with an acyl acceptor M4 or D44 leading to LD-TPase activity.

We have included a sentence in line with the reviewer's recommendation (lines 184-188).

L210-211: the authors indicate that they tested LDT_{Go} activity on muramidase-digested peptidoglycan or purified muropeptides without success. Which experiments were performed? Was the activity tested with non-reduced muropeptides? Reduction of muropeptides with sodium borohydride could inhibit enzymatic activity.

The reviewer raises a valid point. We tested both LDT_{Go} and LDT_{Bcn} activities *in vitro* under different conditions and using different purified substrates such as M2, M4, M5 and their combinations. To rule out a possible effect of reduction of the muropeptides on the enzymatic activity, we also used non-reduced mutanolysin digests of different sacculi such as that of *Gluconobacter oxydans*, *Escherichia coli*, *Vibrio cholerae*. These *in vitro* assays were done using different reaction buffer conditions, including a range of pH (pH 6-8), and supplementation with different salts and metal ions (NaCl, CaCl₂, ZnCl₂, MgCl₂). Unfortunately, all our efforts to demonstrate the activity on purified muropeptides were unsuccessful and have led us to conclude the enzyme is preferentially active on intact sacculi.

Fig. S9: Incorporation of Gln in M2 by exchange with D-Glu seems to be independent of LDT_{Go}, since it is present in the deletion mutant. Consequently, the extraction ion chromatogram shown for M2Gln shown in Fig. 4D could be removed because it does not support the D,L-aminoacid exchange activity of LDT_{Go}.

We agree with the reviewer that the incorporation of Gln is independent of LDT_{Go}; however, we prefer to keep this information in the figure as it includes the identification of non-canonical M2 muropeptides. Nonetheless, we have edited the text to emphasize M2^{Gln} is not a product of LDT_{Go} (lines 236-238).

Fig. S8: Similar suggestion for the MS spectra of M2Gln shown in Fig. S8.

We appreciate the reviewer's suggestion. Please, see our response above.

Minor comments:

L173: change "...upon the expression of LDT_{Bcn} " to "...upon the expression of LDT_{Bcn} in Δ ldtGo mutant (Fig. 1D)".

The quantification referred to upon expression of LDT_{Bcn} is in *E. coli*, not *G. oxydans* Δ/dt_{Go} mutant. We have updated the manuscript to clarify this (line 172).

L194: change to “the L,D-TPase LdtA”

Modified accordingly in the revised version of the manuscript (line 196).

L204: change to “ by an LD-carboxypeptidase (LdcA)”

Modified accordingly in the revised version of the manuscript (line 207).

L536: indicate the OD of overnight cultures.

The OD of both exponential phase and overnight cultures have been included (lines 556-557).

L549: change to .. “N-acetyl-muramic acid residues were reduced to muramicitol”.

Modified accordingly in the revised version of the manuscript (line 570).

L563: does “MS-MS/MS” mean “MS and MS/MS”?

Yes, the reviewer is right, so we have changed it in the revised manuscript (line 585).

L565: define MSE

According to the reviewer’s suggestion, we have defined what MS^E is in the text (lines 587-588).

L578: how were the sacculi prepared?

We have renamed the method as “Sacculi and muropeptides preparation” and clarified the method for sacculi preparation and further digestion with mutanolysin for preparation of muropeptides (lines 564-566).

L581, 584, 586: precise “heat-inactivation”

As requested, we have indicated how heat-inactivation is performed (by boiling for 5 min) (lines 567, 607, 610 and 613).

L586: change “muramidase” to mutanolysin.

We have done the requested replacement (line 566) and throughout the whole manuscript.

Reviewer #2 (Remarks to the Author):

Traditionally, the peptidoglycan (PG) studies were done in model bacteria such as *E. coli*, *Vibrio cholerae* or *Bacillus subtilis* and these yielded a wealth of information on the basic fundamental processes involved in bacterial cell wall biogenesis. PG is a polymer consisting of linear glycan strands attached to a short-stem peptides. In Gram-negative bacteria, the predominant components of the stem peptide include L-alanine (L-Ala1), D-glutamate (D-Glu2), meso-diaminopimelic acid (mDAP3), D-alanine (D-Ala4), and D-Ala5, with cross-links occurring between D-Ala4 and mDAP3 (4-3) or between two mDAP3 residues (3-3). While 4-3 cross-links are made by penicillin-binding proteins, 3-3 cross-links are the products of L,D-Transpeptidases. Recent studies on the PG of non-model culturable bacteria have uncovered novel modifications in the cell wall, providing intriguing insights into their peptidoglycan biology. In the present study, Espaillat et al. describe how the non-canonical 1-3 cross-links in *Glucanobacter oxydans*, a member of alpha-proteobacteria are synthesized and regulated. Through extensive bioinformatics and biochemical approaches, the authors discovered and characterized a new class of enzymes involved in the formation of these 1,3 cross-links, providing evidence of how it differs from its similar counterparts, LDTs (3-3). Additionally, they elucidated the protein structure and offered mechanistic insights into its catalytic activity. This study is truly fascinating presenting clear data and a complete picture. Manuscript is written well with good flow and logic. I thoroughly enjoyed reading the manuscript. This study provides a major breakthrough in the field of peptidoglycan biology and will be of interest to a large class of bacterial research community.

We thank the reviewer for the careful read of our manuscript and their positive comments.

I have no major concerns, but authors should consider the following:

1. Fig. 1A. Representation of T144 needs to be rechecked. Will the central mDAP make 4 bonds which are theoretically impossible? I think the D-ala of central monomer may make a crosslink with mDAP of right monomer.

We thank the reviewer for pointing this out and have fixed the schematics of the T144 in Fig. 1A.

2. Recommend adding a panel to Fig. 1 illustrating all the structures/ denotations of muropeptides including M1, M2 and others (which are already here).

As suggested by the reviewer, we have expanded Fig. 1A to include all the muropeptides indicated in the chromatogram.

3. Fig. 3A. A representation of M2 in the chromatogram is needed for the reader to compare the chromatogram for level of M2. Also label the M2 peak in the chromatogram.

Thanks for pointing this out. We have labelled the M2 peak in the chromatogram in Fig. 3A, but we believe including this muropeptide in the schematics could be misleading as these only illustrate the species that we observe participate in the LD1,3-TPase reaction.

4. In the results section 6 (322–324), it is suggested that three amino acids, Tyr 180, 181, and 182, play crucial role in conferring the characteristic 1-3 cross-linking properties to these enzymes. Have the authors attempted to mutate these residues and assess whether the enzyme still functions as a 1-3 cross-linker?

We are currently working on a follow-up mechanistic study which includes biochemical characterization of LDT_{G60} mutant variants in key residues defined by the structure such as those in the active site or shaping the belt.

[redacted]

5. In Fig. 1, I would suggest adding a western blot showing the expression of catalytic site mutants.

We appreciate the reviewer's suggestion. Unfortunately, the constructs used for complementation in *G. oxydans* lack a tag that would allow immunodetection of the protein. However, all these constructs were done on the same plasmid background and also induction conditions were the same. A Coomassie-stained SDS-PAGE of total protein extract of *G. oxydans* samples shows a band corresponding to LDT_{G60} MW in the complemented samples (Fig. R2). Given that the Pm promoter of the pSEVA238 is leaky in *Gluconobacter*, this band is visible both in non-induced and induced conditions with 1 mM 3-methylbenzoic acid, however it does increase upon addition of the inducer. Further, the intensity of this band is similar for the WT and the catalytic site mutant variants.

Figure R2. Coomassie-stained gel of total protein extract of the *G. oxydans* samples shown in Fig. 1C.

Additionally, the role of the catalytic site mutants in its homologue LDT_{Bcn} is further demonstrated in the heterologous expression in *E. coli* assays shown in Fig. 2C and in the *in vitro* results presented in Fig. 3C and Supplementary Fig. 4 where protein and substrate amounts were the same in all the assays performed (Fig. R3).

Figure R3. Coomassie-stained gel of purified LDT_{Bcn} proteins used in *in vitro* assays shown in Fig. 3C and Supplementary Fig. 4.

6. The figures need to be enlarged and font sizes increased for ease in reading.

We have increased font sizes and enlarged the figures whenever possible.

7. Discussion, lines 420, this paragraph discusses the regulation and homeostasis of different crosslinks. It is possible that 1,3- crosslink hydrolase may exist in these organisms and it may control the level of 1,3 crosslinks. Only the synthases need not be controlled; hydrolases may also contribute to the crosslinkage frequency.

We appreciate the reviewer's comment. We have added a sentence to the discussion as suggested (lines 445-448).

8. May be a sentence on how these crosslinked muopeptides are recycled would help in the Discussion.

We agree with the reviewer that this is an interesting topic. However, *G. oxydans* has no homologues to AmpG or other known mucopeptide transport systems described so far (Gilmore and Cava, Nat Commun 2022, PMID: 36566216). Therefore, we prefer not to speculate and focus on the description and characterization of the novel LD1,3-TPase.

9. Line 54- comma to be included after pentapeptide.

Modified accordingly in the revised version of the manuscript (line 47).

10. Line 87- 'type' after novel can be removed.

Modified accordingly in the revised version of the manuscript (line 80).

Reviewer #3 (Remarks to the Author):

This is an interesting mechanistic study regarding a novel L,D transpeptidase that catalyzes 1-3 stem peptide cross-links. Authors employ extensive bacterial genetics techniques, paired with structural biology and molecular dynamics, to extensively characterize not only the novel transpeptidase itself but also the biochemical and mechanistic environment that lead to its activation. The text is elegantly written and the figures are quite clear. There are a few points that merit clarification, highlighted below.

We thank the reviewer for their positive comments.

Line 20, and elsewhere: authors should include a few sentences to describe the generalized term 'acetic acid bacteria', i.e., specific growth conditions, unusual environments, or metabolism.

We have included a sentence in the Introduction further describing these microbes (lines 80-82).

Lines 116-117: what were the consequences of the 1-3 crosslinks on the overall bacterial phenotype? Did authors observe modifications in growth, shape, or survival capacity? It could be of interest to mention this here.

We appreciate the reviewer's suggestion. These questions are already addressed in lines 386-390 and Fig. 7 and Supplementary Fig. 18. We prefer to reserve the phenotypic characterization until the conclusion of the manuscript to enhance the overall narrative flow.

Lines 142-145: can authors mention a few prototypical strains in these phyla, and are there specific characteristics (such as typical growth environment, pathogenicity potential, etc) that could explain the need for such a transpeptidase in these strains?

As suggested by the reviewer, we have included some representative strains of those phyla that present LDT_{1,3}-like proteins (lines 139-144). Unfortunately, these groups are too heterogenic to allow us drawing any conclusions regarding a need for such type of TPase.

Line 179: kindly clarify the need for Cu⁺² in this experiment.

Cu²⁺ is LD_{3,3}-TPase inhibitor. We have included the missing reference in the manuscript (line 178).

Lines 212-213: can authors comment on the preference of LDT_{Bcn} for PG chains rather than individual muropeptides?

Our efforts to demonstrate the activity of LDT_{Bcn} on purified muropeptides (even using non-reduced muropeptides) were unsuccessful, suggesting this enzyme is preferentially active on intact sacculi. As mutanolysin cleaves the β(1→4) glycosidic bonds of the peptidoglycan releasing free muropeptides, we reasoned this enzyme

might recognize larger structures such as short glycan chains. We have clarified this point further in the text (line 216).

Lines 246-247: it would be of interest to the reader to find out at this point a few more details regarding how the crystal structure was solved. This reviewer had to look for this info in the Materials & Methods section. The structure was solved by using an AlphaFold model, but no details of the experiment were provided, neither in the Results section nor in M&M itself. Can authors comment on differences between the initial model and the final structure, especially in what relates to the belt region and other loops? This is shown in Fig 5 but not explored in the text.

We appreciate the reviewer's suggestion. We have slightly expanded the results explaining this in more detail (lines 256-260).

Differences between the final structure and the AlphaFold2 model are also presented in detail in lines 274-292 and shown in fig 5D and in the Supplementary Fig. 13 and Supplementary Movie 1 and discussed in lines 457-459. The belt region is too disordered to be modelled by AlphaFold2 with accuracy, therefore we cannot compare the model with the final structure. Other relevant loops are further discussed in the afore-mentioned paragraphs.

Minor comments

-line 289, should read 'we subsequently leveraged ...'

Modified accordingly in the revised version of the manuscript (line 302).

- line 607, should read 'Coot was used ...'

Modified accordingly in the revised version of the manuscript (line 634).

-there are a few units missing in the crystallography table, notably in what relates to wavelength, resolution, B-factors, RMS ... in addition, the number of digits to the right of the decimal point could be homogenized.

We thank the reviewer for pointing this out. The Table 1 has been updated accordingly.